# Absence of detectable bovine leukemia virus miRNAs in human cancer small RNA-seq datasets

Juan Pablo Jaworski[1]

**ABSTRACT** Bovine leukemia virus (BLV) encodes 10 mature microRNAs (miRNAs) that are highly expressed in infected cattle and have been implicated in oncogenic mechanisms. A potential association between BLV and human cancer remains controversial: while several reports have detected BLV proviral DNA in human breast tissue, high-throughput sequencing analyses have failed to identify BLV nucleic acids in human tumors. Here, we performed a high-sensitivity screen for BLV miRNAs across 335 publicly available human small RNA-seq data sets, including breast cancer (BCA), acute lymphoblastic leukemia, acute myeloid leukemia, chronic lymphocytic leukemia, and healthy controls, totaling approximately 1.98 billion trimmed reads. To validate our analytical workflow, six small RNA data sets from BLV-infected and uninfected cattle were processed in parallel. Across the human data sets, only 60 reads (0.000003%) aligned to BLV miRNA reference sequences, predominantly STAR strands or sequences containing mismatches—patterns consistent with background noise rather than biologically meaningful viral miRNAs. In contrast, BLV miRNAs were robustly detected across BLV-infected cattle samples, confirming pipeline sensitivity for canonical BLV miRNAs. These results demonstrate that BLV miRNAs are absent or below the operational detection threshold in human BCA and human leukemia miRNomes using standard small-RNA sequencing workflows and canonical reference sequences.

**IMPORTANCE** Bovine leukemia virus (BLV) causes leukemia in cattle. BLV genome encodes for microRNAs (miRNAs) that can influence how cells grow. Some researchers have suggested that BLV might also play a role in human cancers, especially breast cancer (BCA), but scientific studies have produced conflicting results. In our work, we analyzed nearly two billion genetic sequences from 335 human cancer samples—including BCA and several types of leukemia—to look for BLV miRNAs. We found almost no sequences that resembled BLV miRNAs, and those few likely represented background noise rather than true viral signals. In contrast, BLV miRNAs were easily detected in infected cattle samples. Our results indicate that BLV miRNAs are not present at meaningful levels in particular human cancers (i.e., leukemia and BCA).

**KEYWORDS** micro-RNA, miRNA, cancer, human, bovine leukemia virus, BLV, virus

Address correspondence to Juan Pablo Jaworski, jaworski.juan@inta.gob.ar.

The author declares no conflict of interest.

See the funding table on p. 13.

Bovine leukemia virus (BLV) is a deltaretrovirus that causes malignant B-cell lymphoma in up to 10% of infected cattle (1). There are controversies regarding the association between BLV and cancer development in humans (2). In a series of case-control studies, involving samples of breast tissue from the United States ($n = 224$) and Australia ($n = 96$), Buehring et al. reported that the presence of BLV proviral DNA (determined by *in situ* PCR) augmented the risk of developing breast cancer (BCA) in both populations (OR = 3.1 and OR = 4.7, respectively) (3, 4). In addition, several other publications support the association between exposure to BLV and human cancer (5–11). Although BLV-reactive antibodies have been detected in human plasma (12), the

TABLE 1 PRJNA1126751: Small RNA sequencing of exosomes in BCA screening population (human)[a]

| Sample ID | Cancer type | Post trim reads | Total no of reads | blv-miR-B1-3p | blv-miR-B1-5p | blv-miR-B2-3p | blv-miR-B2-5p | blv-miR-B3-3p | blv-miR-B3-5p | blv-miR-B4-3p | blv-miR-B4-5p | blv-miR-B5-3p | blv-miR-B5-5p | hsa-miR-106b-5p | hsa-miR-21-5p | hsa-mir-29a-3p |
|---|---|---|---|---|---|---|---|---|---|---|---|---|---|---|---|---|
| | | | | | | | | | | | | | | Number of reads that align on | | |
| SRR29496163 | NEGATIVE | 9.3M | 15.0M | 0.00 | 0.00 | 0.00 | 0.00 | 0.00 | 0.00 | 0.00 | 0.00 | 0.00 | 0.00 | 69.00 | 11,786.00 | 5331.00 |
| SRR29496164 | NEGATIVE | 8.4M | 13.4M | 0.00 | 0.00 | 0.00 | 0.00 | 0.00 | 0.00 | 0.00 | 0.00 | 0.00 | 0.00 | 29.00 | 6383.00 | 2001.00 |
| SRR29496165 | NEGATIVE | 9.6M | 15.1M | 0.00 | 0.00 | 0.00 | 0.00 | 0.00 | 0.00 | 0.00 | 0.00 | 0.00 | 0.00 | 29.00 | 12,460.00 | 3044.00 |
| SRR29496166 | NEGATIVE | 8.3M | 13.8M | 0.00 | 1.00 | 0.00 | 0.00 | 0.00 | 0.00 | 0.00 | 0.00 | 0.00 | 0.00 | 9.00 | 3219.00 | 859.00 |
| SRR29496167 | NEGATIVE | 6.7M | 10.6M | 0.00 | 0.00 | 0.00 | 0.00 | 0.00 | 0.00 | 0.00 | 0.00 | 0.00 | 0.00 | 19.00 | 4761.00 | 2493.00 |
| SRR29496168 | NEGATIVE | 7.8M | 12.5M | 0.00 | 0.00 | 0.00 | 0.00 | 7.00 | 0.00 | 0.00 | 0.00 | 0.00 | 0.00 | 40.00 | 8136.00 | 2598.00 |
| SRR29496169 | NEGATIVE | 7.7M | 12.4M | 0.00 | 0.00 | 0.00 | 0.00 | 0.00 | 0.00 | 0.00 | 0.00 | 0.00 | 0.00 | 23.00 | 8055.00 | 2020.00 |
| SRR29496170 | NEGATIVE | 8.3M | 13.5M | 0.00 | 0.00 | 0.00 | 0.00 | 0.00 | 0.00 | 0.00 | 0.00 | 0.00 | 0.00 | 22.00 | 7496.00 | 3312.00 |
| SRR29496171 | NEGATIVE | 7.8M | 12.7M | 0.00 | 0.00 | 0.00 | 0.00 | 0.00 | 0.00 | 0.00 | 1.00 | 0.00 | 0.00 | 45.00 | 8004.00 | 2793.00 |
| SRR29496172 | NEGATIVE | 10.9M | 17.8M | 0.00 | 0.00 | 0.00 | 0.00 | 0.00 | 0.00 | 0.00 | 0.00 | 0.00 | 0.00 | 17.00 | 4379.00 | 1665.00 |
| SRR29496173 | NEGATIVE | 8.4M | 13.0M | 0.00 | 0.00 | 0.00 | 0.00 | 0.00 | 0.00 | 0.00 | 0.00 | 0.00 | 0.00 | 77.00 | 15,023.00 | 6791.00 |
| SRR29496174 | NEGATIVE | 8.4M | 13.9M | 0.00 | 0.00 | 0.00 | 0.00 | 0.00 | 0.00 | 0.00 | 0.00 | 0.00 | 0.00 | 31.00 | 8301.00 | 3669.00 |
| SRR29496175 | NEGATIVE | 7.5M | 12.1M | 0.00 | 0.00 | 0.00 | 0.00 | 0.00 | 0.00 | 0.00 | 0.00 | 0.00 | 0.00 | 53.00 | 17,808.00 | 5983.00 |
| SRR29496176 | NEGATIVE | 7.7M | 12.4M | 0.00 | 0.00 | 0.00 | 0.00 | 0.00 | 0.00 | 0.00 | 0.00 | 0.00 | 0.00 | 32.00 | 12,217.00 | 2788.00 |
| SRR29496177 | NEGATIVE | 9.0M | 14.3M | 0.00 | 0.00 | 0.00 | 0.00 | 0.00 | 0.00 | 0.00 | 0.00 | 0.00 | 0.00 | 36.00 | 9733.00 | 4463.00 |
| SRR29496178 | NEGATIVE | 13.0M | 19.5M | 0.00 | 0.00 | 0.00 | 0.00 | 0.00 | 0.00 | 0.00 | 0.00 | 0.00 | 0.00 | 13.00 | 10,358.00 | 1144.00 |
| SRR29496179 | NEGATIVE | 9.4M | 14.7M | 0.00 | 0.00 | 0.00 | 0.00 | 0.00 | 0.00 | 0.00 | 0.00 | 0.00 | 0.00 | 49.00 | 13,026.00 | 9079.00 |
| SRR29496180 | NEGATIVE | 9.2M | 14.4M | 0.00 | 0.00 | 0.00 | 0.00 | 0.00 | 0.00 | 0.00 | 0.00 | 0.00 | 0.00 | 61.00 | 13,765.00 | 3568.00 |
| SRR29496181 | NEGATIVE | 8.6M | 14.5M | 0.00 | 0.00 | 0.00 | 0.00 | 0.00 | 0.00 | 0.00 | 0.00 | 0.00 | 0.00 | 18.00 | 2755.00 | 1067.00 |
| SRR29496182 | NEGATIVE | 8.9M | 14.6M | 0.00 | 0.00 | 0.00 | 0.00 | 0.00 | 0.00 | 0.00 | 0.00 | 0.00 | 0.00 | 23.00 | 7847.00 | 1307.00 |
| SRR29496183 | NEGATIVE | 6.9M | 11.2M | 0.00 | 0.00 | 0.00 | 0.00 | 0.00 | 0.00 | 0.00 | 0.00 | 0.00 | 2.00 | 11.00 | 3005.00 | 1461.00 |
| SRR29496184 | NEGATIVE | 10.2M | 15.5M | 0.00 | 0.00 | 0.00 | 0.00 | 0.00 | 0.00 | 0.00 | 0.00 | 0.00 | 0.00 | 145.00 | 21,714.00 | 5540.00 |
| SRR29496185 | NEGATIVE | 8.8M | 15.1M | 0.00 | 0.00 | 0.00 | 0.00 | 0.00 | 0.00 | 0.00 | 0.00 | 0.00 | 0.00 | 25.00 | 9108.00 | 3620.00 |
| SRR29496186 | NEGATIVE | 6.8M | 12.5M | 0.00 | 0.00 | 0.00 | 0.00 | 0.00 | 0.00 | 0.00 | 0.00 | 0.00 | 0.00 | 32.00 | 11,560.00 | 3345.00 |
| SRR29496247 | NEGATIVE | 9.3M | 15.9M | 0.00 | 0.00 | 0.00 | 0.00 | 0.00 | 0.00 | 0.00 | 0.00 | 0.00 | 0.00 | 44.00 | 5744.00 | 1832.00 |
| SRR29496248 | NEGATIVE | 7.3M | 12.5M | 0.00 | 0.00 | 0.00 | 0.00 | 0.00 | 0.00 | 0.00 | 0.00 | 0.00 | 0.00 | 36.00 | 6364.00 | 2662.00 |
| SRR29496249 | NEGATIVE | 10.8M | 16.0M | 0.00 | 0.00 | 0.00 | 0.00 | 0.00 | 0.00 | 0.00 | 0.00 | 0.00 | 0.00 | 88.00 | 14,772.00 | 6262.00 |
| SRR29496250 | NEGATIVE | 9.6M | 15.8M | 0.00 | 0.00 | 0.00 | 0.00 | 0.00 | 0.00 | 0.00 | 1.00 | 0.00 | 0.00 | 24.00 | 4167.00 | 1985.00 |
| SRR29496251 | NEGATIVE | 7.6M | 12.2M | 0.00 | 0.00 | 0.00 | 0.00 | 0.00 | 0.00 | 0.00 | 0.00 | 0.00 | 0.00 | 22.00 | 3326.00 | 1173.00 |
| SRR29496252 | NEGATIVE | 9.1M | 14.1M | 0.00 | 0.00 | 0.00 | 0.00 | 0.00 | 0.00 | 0.00 | 0.00 | 0.00 | 0.00 | 45.00 | 14,499.00 | 5077.00 |
| SRR29496253 | NEGATIVE | 7.6M | 12.2M | 0.00 | 0.00 | 0.00 | 0.00 | 0.00 | 0.00 | 0.00 | 0.00 | 0.00 | 3.00 | 33.00 | 5675.00 | 2108.00 |
| SRR29496254 | NEGATIVE | 7.8M | 12.2M | 0.00 | 0.00 | 0.00 | 0.00 | 0.00 | 0.00 | 0.00 | 2.00 | 0.00 | 0.00 | 42.00 | 6571.00 | 4075.00 |
| SRR29496255 | NEGATIVE | 11.6M | 18.2M | 0.00 | 0.00 | 0.00 | 0.00 | 0.00 | 0.00 | 0.00 | 0.00 | 0.00 | 0.00 | 30.00 | 9226.00 | 5788.00 |
| SRR29496256 | NEGATIVE | 7.6M | 12.2M | 0.00 | 0.00 | 0.00 | 0.00 | 0.00 | 0.00 | 0.00 | 0.00 | 0.00 | 0.00 | 53.00 | 11,636.00 | 7966.00 |
| SRR29496257 | NEGATIVE | 9.9M | 15.2M | 0.00 | 0.00 | 0.00 | 0.00 | 0.00 | 0.00 | 0.00 | 0.00 | 0.00 | 0.00 | 80.00 | 17,320.00 | 6749.00 |
| SRR29496258 | NEGATIVE | 8.2M | 13.4M | 0.00 | 0.00 | 0.00 | 0.00 | 0.00 | 0.00 | 0.00 | 0.00 | 0.00 | 0.00 | 50.00 | 11,005.00 | 2757.00 |
| SRR29496259 | NEGATIVE | 8.6M | 15.0M | 0.00 | 0.00 | 0.00 | 0.00 | 0.00 | 0.00 | 0.00 | 0.00 | 0.00 | 0.00 | 20.00 | 9087.00 | 2386.00 |
| SRR29496291 | NEGATIVE | 8.6M | 14.0M | 0.00 | 0.00 | 0.00 | 0.00 | 0.00 | 0.00 | 0.00 | 0.00 | 0.00 | 0.00 | 20.00 | 5939.00 | 2722.00 |
| SRR29496292 | NEGATIVE | 7.4M | 12.9M | 0.00 | 0.00 | 0.00 | 0.00 | 0.00 | 0.00 | 0.00 | 0.00 | 0.00 | 0.00 | 56.00 | 6018.00 | 3140.00 |
| SRR29496293 | NEGATIVE | 8.5M | 14.1M | 0.00 | 0.00 | 0.00 | 0.00 | 0.00 | 0.00 | 0.00 | 0.00 | 0.00 | 0.00 | 58.00 | 21,033.00 | 3977.00 |
| SRR29496294 | NEGATIVE | 8.6M | 13.9M | 0.00 | 1.00 | 0.00 | 1.00 | 0.00 | 0.00 | 0.00 | 0.00 | 0.00 | 0.00 | 38.00 | 6263.00 | 1745.00 |
| SRR29496295 | NEGATIVE | 7.8M | 12.6M | 0.00 | 0.00 | 0.00 | 0.00 | 0.00 | 0.00 | 0.00 | 0.00 | 0.00 | 0.00 | 39.00 | 6926.00 | 1472.00 |
| SRR29496296 | NEGATIVE | 10.0M | 15.7M | 0.00 | 0.00 | 0.00 | 0.00 | 0.00 | 0.00 | 0.00 | 0.00 | 0.00 | 0.00 | 44.00 | 10,016.00 | 4194.00 |
| SRR29496297 | NEGATIVE | 9.5M | 14.2M | 0.00 | 0.00 | 0.00 | 0.00 | 0.00 | 0.00 | 0.00 | 0.00 | 0.00 | 0.00 | 69.00 | 33,096.00 | 7215.00 |
| SRR29496298 | NEGATIVE | 7.6M | 12.0M | 0.00 | 0.00 | 0.00 | 0.00 | 0.00 | 0.00 | 0.00 | 0.00 | 0.00 | 0.00 | 28.00 | 6102.00 | 2521.00 |
| SRR29496299 | NEGATIVE | 7.7M | 13.5M | 0.00 | 0.00 | 0.00 | 0.00 | 0.00 | 0.00 | 0.00 | 0.00 | 0.00 | 0.00 | 39.00 | 7521.00 | 2868.00 |
| SRR29496300 | NEGATIVE | 8.3M | 14.6M | 0.00 | 0.00 | 0.00 | 0.00 | 0.00 | 0.00 | 0.00 | 0.00 | 0.00 | 0.00 | 127.00 | 19,201.00 | 9511.00 |
| SRR29496301 | NEGATIVE | 10.7M | 19.8M | 0.00 | 1.00 | 0.00 | 0.00 | 0.00 | 0.00 | 0.00 | 0.00 | 0.00 | 0.00 | 47.00 | 17,131.00 | 4744.00 |
| SRR29496302 | NEGATIVE | 6.7M | 11.1M | 0.00 | 0.00 | 0.00 | 0.00 | 0.00 | 0.00 | 0.00 | 0.00 | 0.00 | 0.00 | 25.00 | 3845.00 | 1369.00 |

TABLE 1 [PRJNA1126751](): Small RNA sequencing of exosomes in BCA screening population (human)[a] (*Continued*)

| Sample ID | Cancer type | Post trim reads | Total no of reads | blv-miR-B1-3p | blv-miR-B1-5p | blv-miR-B2-3p | blv-miR-B2-5p | blv-miR-B3-3p | blv-miR-B3-5p | blv-miR-B4-3p | blv-miR-B4-5p | blv-miR-B5-3p | blv-miR-B5-5p | hsa-miR-106b-5p | hsa-miR-21-5p | hsa-mir-29a-3p |
|---|---|---|---|---|---|---|---|---|---|---|---|---|---|---|---|---|
| | | | | | | | | | | | | | | Number of reads that align on | | |
| SRR29496303 | NEGATIVE | 6.9M | 13.5M | 0.00 | 0.00 | 0.00 | 0.00 | 0.00 | 0.00 | 0.00 | 0.00 | 0.00 | 0.00 | 26.00 | 3453.00 | 3652.00 |
| SRR29496304 | NEGATIVE | 4.8M | 11.9M | 0.00 | 0.00 | 0.00 | 0.00 | 0.00 | 0.00 | 0.00 | 0.00 | 0.00 | 0.00 | 202.00 | 11,127.00 | 7007.00 |
| SRR29496305 | NEGATIVE | 7.7M | 14.4M | 0.00 | 0.00 | 0.00 | 0.00 | 0.00 | 0.00 | 0.00 | 0.00 | 0.00 | 0.00 | 20.00 | 8072.00 | 4234.00 |
| SRR29496306 | NEGATIVE | 2.3M | 3.9M | 0.00 | 0.00 | 0.00 | 0.00 | 0.00 | 0.00 | 0.00 | 0.00 | 0.00 | 0.00 | 6.00 | 648.00 | 384.00 |
| SRR29496307 | NEGATIVE | 4.6M | 7.8M | 0.00 | 0.00 | 0.00 | 0.00 | 0.00 | 0.00 | 0.00 | 0.00 | 0.00 | 0.00 | 3.00 | 1243.00 | 688.00 |
| SRR29496308 | NEGATIVE | 5.7M | 9.9M | 0.00 | 0.00 | 0.00 | 0.00 | 0.00 | 0.00 | 0.00 | 3.00 | 0.00 | 0.00 | 14.00 | 3290.00 | 1718.00 |
| SRR29496309 | NEGATIVE | 5.2M | 9.9M | 0.00 | 0.00 | 0.00 | 0.00 | 0.00 | 0.00 | 0.00 | 0.00 | 0.00 | 0.00 | 33.00 | 5334.00 | 3610.00 |
| SRR29496310 | NEGATIVE | 5.6M | 10.4M | 0.00 | 0.00 | 0.00 | 0.00 | 0.00 | 0.00 | 0.00 | 3.00 | 0.00 | 0.00 | 26.00 | 3112.00 | 2289.00 |
| SRR29496311 | NEGATIVE | 6.1M | 10.7M | 0.00 | 0.00 | 0.00 | 0.00 | 0.00 | 0.00 | 0.00 | 0.00 | 0.00 | 0.00 | 23.00 | 3111.00 | 2144.00 |
| SRR29496312 | NEGATIVE | 6.2M | 9.3M | 0.00 | 0.00 | 0.00 | 0.00 | 0.00 | 0.00 | 0.00 | 0.00 | 0.00 | 0.00 | 662.00 | 8506.00 | 5711.00 |
| SRR29496313 | NEGATIVE | 4.5M | 9.0M | 0.00 | 0.00 | 0.00 | 0.00 | 0.00 | 0.00 | 0.00 | 0.00 | 0.00 | 0.00 | 11.00 | 5996.00 | 2144.00 |
| SRR29496314 | NEGATIVE | 8.7M | 14.9M | 0.00 | 0.00 | 0.00 | 0.00 | 0.00 | 0.00 | 0.00 | 0.00 | 0.00 | 0.00 | 18.00 | 9444.00 | 1809.00 |
| SRR29496191 | BCA | 5.2M | 9.9M | 0.00 | 0.00 | 0.00 | 0.00 | 0.00 | 0.00 | 0.00 | 2.00 | 0.00 | 0.00 | 13.00 | 5060.00 | 3922.00 |
| SRR29496196 | BCA | 5.3M | 9.8M | 0.00 | 0.00 | 0.00 | 0.00 | 0.00 | 0.00 | 0.00 | 0.00 | 0.00 | 0.00 | 56.00 | 10,323.00 | 4574.00 |
| SRR29496197 | BCA | 5.4M | 10.6M | 0.00 | 0.00 | 0.00 | 0.00 | 0.00 | 0.00 | 0.00 | 0.00 | 0.00 | 0.00 | 7.00 | 4455.00 | 2074.00 |
| SRR29496202 | BCA | 6.1M | 11.8M | 0.00 | 0.00 | 0.00 | 0.00 | 0.00 | 0.00 | 0.00 | 0.00 | 0.00 | 0.00 | 57.00 | 4946.00 | 4934.00 |
| SRR29496205 | BCA | 6.4M | 10.5M | 0.00 | 0.00 | 0.00 | 0.00 | 0.00 | 0.00 | 0.00 | 0.00 | 0.00 | 0.00 | 27.00 | 2975.00 | 1934.00 |
| SRR29496212 | BCA | 8.8M | 13.7M | 0.00 | 0.00 | 0.00 | 0.00 | 0.00 | 0.00 | 0.00 | 0.00 | 0.00 | 0.00 | 181.00 | 10,771.00 | 7760.00 |
| SRR29496215 | BCA | 7.1M | 11.3M | 0.00 | 0.00 | 0.00 | 0.00 | 0.00 | 0.00 | 0.00 | 0.00 | 0.00 | 0.00 | 56.00 | 14,010.00 | 6758.00 |

[a]BCA: breast cancer.

potential route of BLV infection in humans is not known. In this regard, the consumption of untreated dairy products from BLV-infected animals has been proposed as a possible one. Another publication by Dr. Buehring reported the presence of BLV in human blood (13). Oppositely, one large study failed to demonstrate any association between BLV and BCA in Chinese patients using different serological and molecular assays (14). Additionally, whole-genome and whole-transcriptome sequencing analysis of BCA samples, obtained from The Cancer Genome Atlas (TCGI, National Center for Biotechnology Information [NCBI]), failed to detect sequencing reads corresponding to proviral DNA or viral transcripts in these samples (15, 16). These conflicting findings prompted interest in exploring alternative viral factors, such as BLV-derived microRNAs (miRNAs), that might contribute to oncogenic processes.

miRNAs are small RNA (sRNA) molecules, typically 19–24 nucleotides long, that bind by complementarity with sequences present in messenger RNAs (mRNA). As a result of this interaction, the target mRNA is degraded, and translation is suppressed (17). It is widely assumed that miRNAs are constitutively dysregulated in different human cancer types, affecting key biological pathways that finally lead to tumor progression (18). Recently, BLV-encoded miRNAs have been associated with the regulation of different genes involved in oncogenic pathways. One of BLV-encoded miRNAs (blv-miR-b4-3p) shares nine nucleotides of its seed region with a miR-29a (19, 20). miR-29a is a well-described oncomiR that targets HMG-box transcription factor 1 (HBP-1) and peroxidasin homolog (PXDN) mRNAs, reducing their expression (19, 20). Both HBP-1 and PXDN gene products have anti-tumoral activity in B cells. Endogenous miR-29a overexpression is linked to B-cell malignancies in humans (21). *In vitro* and *ex vivo* studies showed that blv-miR-b4-3p downregulated the expression of HBP-1 and PXDN (19, 20), and *in vivo* expression of blv-miR-b4-3p was linked to PXDN downregulation in cattle naturally infected with BLV (22). Considering the oncogenic role of miR-29a, it is possible that blv-miR-b4-3p could be involved in BLV-associated tumorigenesis mechanisms.

It is also known that miRNAs can be conveyed through lipid extracellular vesicles (EVs) present in milk and colostrum of humans and cattle (23–25). These miRNAs are refractory

**TABLE 2** PRJNA996975: Profiling of miRNA in pediatric AML in the Egyptian population[a]

| Sample ID | Cancer type | Post trim reads | Total no of reads | blv-miR-B1-3p | blv-miR-B1-5p | blv-miR-B2-3p | blv-miR-B2-5p | blv-miR-B3-3p | blv-miR-B3-5p | blv-miR-B4-3p | blv-miR-B4-5p | blv-miR-B5-3p | blv-miR-B5-5p | hsa-miR-106b-5p | hsa-miR-21-5p | hsa-mir-29a-3p |
|---|---|---|---|---|---|---|---|---|---|---|---|---|---|---|---|---|
| SRR25374631 | NEGATIVE | 1.2M | 1.2M | 0.00 | 0.00 | 0.00 | 0.00 | 0.00 | 0.00 | 0.00 | 0.00 | 0.00 | 0.00 | 0.00 | 1.00 | 0.00 |
| SRR25374632 | NEGATIVE | 0.3M | 0.3M | 0.00 | 0.00 | 0.00 | 0.00 | 0.00 | 0.00 | 0.00 | 0.00 | 0.00 | 0.00 | 1.00 | 0.00 | 0.00 |
| SRR25374633 | NEGATIVE | 1.5M | 1.8M | 0.00 | 0.00 | 0.00 | 0.00 | 0.00 | 0.00 | 0.00 | 0.00 | 0.00 | 0.00 | 0.00 | 3.00 | 0.00 |
| SRR25374634 | AML | 1.3M | 1.3M | 0.00 | 0.00 | 0.00 | 0.00 | 0.00 | 0.00 | 0.00 | 0.00 | 0.00 | 0.00 | 4.00 | 63.00 | 1.00 |
| SRR25374635 | AML | 1.2M | 1.2M | 0.00 | 0.00 | 0.00 | 0.00 | 0.00 | 0.00 | 0.00 | 0.00 | 0.00 | 0.00 | 3.00 | 70.00 | 0.00 |
| SRR25374636 | AML | 1.0M | 1.0M | 0.00 | 0.00 | 0.00 | 0.00 | 0.00 | 0.00 | 0.00 | 0.00 | 0.00 | 0.00 | 3.00 | 44.00 | 0.00 |
| SRR25374637 | AML | 0.7M | 0.8M | 0.00 | 0.00 | 0.00 | 0.00 | 0.00 | 0.00 | 0.00 | 0.00 | 0.00 | 0.00 | 3.00 | 7.00 | 0.00 |
| SRR25374638 | AML | 0.8M | 0.9M | 0.00 | 0.00 | 0.00 | 0.00 | 0.00 | 0.00 | 0.00 | 0.00 | 0.00 | 0.00 | 3.00 | 22.00 | 0.00 |
| SRR25374639 | AML | 1.0M | 1.1M | 0.00 | 0.00 | 0.00 | 0.00 | 0.00 | 0.00 | 0.00 | 0.00 | 0.00 | 0.00 | 0.00 | 95.00 | 0.00 |
| SRR25374640 | AML | 0.8M | 0.9M | 0.00 | 0.00 | 0.00 | 0.00 | 0.00 | 0.00 | 0.00 | 0.00 | 0.00 | 0.00 | 0.00 | 11.00 | 0.00 |
| SRR25374641 | AML | 0.6M | 0.6M | 0.00 | 0.00 | 0.00 | 0.00 | 0.00 | 0.00 | 0.00 | 0.00 | 0.00 | 0.00 | 0.00 | 19.00 | 0.00 |
| SRR25374642 | AML | 0.8M | 0.9M | 0.00 | 0.00 | 0.00 | 0.00 | 0.00 | 0.00 | 0.00 | 0.00 | 0.00 | 0.00 | 1.00 | 14.00 | 0.00 |
| SRR25374643 | AML | 1.0M | 1.0M | 0.00 | 0.00 | 0.00 | 0.00 | 0.00 | 0.00 | 0.00 | 0.00 | 0.00 | 0.00 | 1.00 | 28.00 | 0.00 |
| SRR25374644 | AML | 1.0M | 1.1M | 0.00 | 0.00 | 0.00 | 0.00 | 0.00 | 0.00 | 0.00 | 0.00 | 0.00 | 0.00 | 6.00 | 148.00 | 1.00 |
| SRR25374645 | AML | 1.1M | 1.2M | 0.00 | 0.00 | 0.00 | 0.00 | 0.00 | 0.00 | 0.00 | 0.00 | 0.00 | 0.00 | 1.00 | 108.00 | 0.00 |
| SRR25374646 | AML | 0.8M | 0.8M | 0.00 | 0.00 | 0.00 | 0.00 | 0.00 | 0.00 | 0.00 | 0.00 | 0.00 | 0.00 | 5.00 | 9.00 | 0.00 |
| SRR25374647 | AML | 0.8M | 0.9M | 0.00 | 0.00 | 0.00 | 0.00 | 0.00 | 0.00 | 0.00 | 0.00 | 0.00 | 0.00 | 0.00 | 35.00 | 0.00 |
| SRR25374648 | AML | 0.9M | 1.0M | 0.00 | 0.00 | 0.00 | 0.00 | 0.00 | 0.00 | 0.00 | 0.00 | 0.00 | 0.00 | 0.00 | 29.00 | 0.00 |
| SRR25374649 | AML | 1.2M | 1.2M | 0.00 | 0.00 | 0.00 | 0.00 | 0.00 | 0.00 | 0.00 | 0.00 | 0.00 | 0.00 | 1.00 | 77.00 | 0.00 |
| SRR25374650 | AML | 0.9M | 0.9M | 0.00 | 0.00 | 0.00 | 0.00 | 0.00 | 0.00 | 0.00 | 0.00 | 0.00 | 0.00 | 6.00 | 44.00 | 0.00 |
| SRR25374651 | AML | 0.3M | 0.3M | 0.00 | 0.00 | 0.00 | 0.00 | 0.00 | 0.00 | 0.00 | 0.00 | 0.00 | 0.00 | 0.00 | 1.00 | 0.00 |
| SRR25374652 | AML | 0.3M | 0.3M | 0.00 | 0.00 | 0.00 | 0.00 | 0.00 | 0.00 | 0.00 | 0.00 | 0.00 | 0.00 | 1.00 | 17.00 | 0.00 |
| SRR25374653 | AML | 0.8M | 0.8M | 0.00 | 0.00 | 0.00 | 0.00 | 0.00 | 0.00 | 0.00 | 0.00 | 0.00 | 0.00 | 1.00 | 27.00 | 0.00 |
| SRR25374654 | AML | 0.9M | 0.9M | 0.00 | 0.00 | 0.00 | 0.00 | 0.00 | 0.00 | 0.00 | 0.00 | 0.00 | 0.00 | 0.00 | 11.00 | 0.00 |
| SRR25374655 | AML | 1.0M | 1.0M | 0.00 | 0.00 | 0.00 | 0.00 | 0.00 | 0.00 | 0.00 | 0.00 | 0.00 | 0.00 | 1.00 | 10.00 | 0.00 |
| SRR25374656 | AML | 0.9M | 0.9M | 0.00 | 0.00 | 0.00 | 0.00 | 0.00 | 0.00 | 0.00 | 0.00 | 0.00 | 0.00 | 2.00 | 65.00 | 0.00 |
| SRR25374657 | AML | 0.6M | 0.7M | 0.00 | 0.00 | 0.00 | 0.00 | 0.00 | 0.00 | 0.00 | 0.00 | 0.00 | 0.00 | 1.00 | 61.00 | 0.00 |
| SRR25374658 | AML | 0.7M | 0.7M | 0.00 | 0.00 | 0.00 | 0.00 | 0.00 | 0.00 | 0.00 | 0.00 | 0.00 | 0.00 | 1.00 | 4.00 | 0.00 |
| SRR25374659 | AML | 0.9M | 0.9M | 0.00 | 0.00 | 0.00 | 0.00 | 0.00 | 0.00 | 0.00 | 0.00 | 0.00 | 0.00 | 0.00 | 68.00 | 0.00 |
| SRR25374660 | AML | 0.7M | 0.7M | 0.00 | 0.00 | 0.00 | 0.00 | 0.00 | 0.00 | 0.00 | 0.00 | 0.00 | 0.00 | 6.00 | 175.00 | 0.00 |
| SRR25374661 | AML | 0.9M | 0.9M | 0.00 | 0.00 | 0.00 | 0.00 | 0.00 | 0.00 | 0.00 | 0.00 | 0.00 | 0.00 | 2.00 | 28.00 | 0.00 |
| SRR25374662 | AML | 0.8M | 0.9M | 0.00 | 0.00 | 0.00 | 0.00 | 0.00 | 0.00 | 0.00 | 1.00 | 0.00 | 0.00 | 4.00 | 75.00 | 0.00 |
| SRR25374663 | AML | 1.4M | 1.6M | 0.00 | 0.00 | 0.00 | 0.00 | 0.00 | 0.00 | 0.00 | 0.00 | 0.00 | 0.00 | 7.00 | 261.00 | 0.00 |
| SRR25374664 | AML | 0.9M | 1.0M | 0.00 | 0.00 | 0.00 | 0.00 | 0.00 | 0.00 | 0.00 | 0.00 | 0.00 | 0.00 | 2.00 | 14.00 | 0.00 |
| SRR25374665 | AML | 1.3M | 1.4M | 0.00 | 0.00 | 0.00 | 0.00 | 0.00 | 0.00 | 0.00 | 0.00 | 0.00 | 0.00 | 7.00 | 62.00 | 0.00 |
| SRR25374666 | AML | 0.8M | 0.8M | 0.00 | 0.00 | 0.00 | 0.00 | 0.00 | 0.00 | 0.00 | 0.00 | 0.00 | 0.00 | 3.00 | 31.00 | 0.00 |
| SRR25374667 | AML | 1.6M | 1.7M | 0.00 | 0.00 | 0.00 | 0.00 | 0.00 | 0.00 | 0.00 | 0.00 | 0.00 | 0.00 | 3.00 | 84.00 | 0.00 |
| SRR25374668 | AML | 0.5M | 0.6M | 0.00 | 0.00 | 0.00 | 0.00 | 0.00 | 0.00 | 0.00 | 0.00 | 0.00 | 0.00 | 0.00 | 4.00 | 0.00 |
| SRR25374669 | AML | 1.5M | 1.6M | 0.00 | 0.00 | 0.00 | 0.00 | 0.00 | 0.00 | 0.00 | 0.00 | 0.00 | 0.00 | 8.00 | 163.00 | 0.00 |
| SRR25374670 | AML | 1.3M | 1.4M | 0.00 | 0.00 | 0.00 | 0.00 | 0.00 | 0.00 | 0.00 | 1.00 | 0.00 | 0.00 | 0.00 | 16.00 | 0.00 |
| SRR25374671 | AML | 1.5M | 1.5M | 0.00 | 0.00 | 0.00 | 0.00 | 0.00 | 0.00 | 0.00 | 0.00 | 0.00 | 0.00 | 4.00 | 23.00 | 0.00 |
| SRR25374672 | AML | 1.5M | 1.6M | 0.00 | 0.00 | 0.00 | 0.00 | 0.00 | 0.00 | 0.00 | 0.00 | 0.00 | 0.00 | 5.00 | 267.00 | 0.00 |
| SRR25374673 | AML | 0.9M | 1.0M | 0.00 | 0.00 | 0.00 | 0.00 | 0.00 | 0.00 | 0.00 | 0.00 | 0.00 | 0.00 | 2.00 | 30.00 | 0.00 |
| SRR25374674 | AML | 0.6M | 0.6M | 0.00 | 0.00 | 0.00 | 0.00 | 0.00 | 0.00 | 0.00 | 0.00 | 0.00 | 0.00 | 0.00 | 9.00 | 0.00 |
| SRR25374675 | AML | 0.8M | 0.8M | 0.00 | 0.00 | 0.00 | 0.00 | 0.00 | 0.00 | 0.00 | 0.00 | 0.00 | 0.00 | 6.00 | 17.00 | 0.00 |
| SRR25374676 | AML | 0.6M | 0.8M | 0.00 | 0.00 | 0.00 | 0.00 | 0.00 | 0.00 | 0.00 | 0.00 | 0.00 | 0.00 | 3.00 | 67.00 | 0.00 |
| SRR25374677 | AML | 0.7M | 0.8M | 0.00 | 0.00 | 0.00 | 0.00 | 0.00 | 0.00 | 0.00 | 0.00 | 0.00 | 0.00 | 0.00 | 116.00 | 0.00 |
| SRR25374678 | AML | 1.1M | 1.1M | 0.00 | 0.00 | 0.00 | 0.00 | 0.00 | 0.00 | 0.00 | 0.00 | 0.00 | 0.00 | 5.00 | 50.00 | 0.00 |

*(Continued on next page)*

**TABLE 2** PRJNA996975: Profiling of miRNA in pediatric AML in the Egyptian population[a] (*Continued*)

| Sample ID | Cancer type | Post trim reads | Total no of reads | blv-miR-B1-3p | blv-miR-B1-5p | blv-miR-B2-3p | blv-miR-B2-5p | blv-miR-B3-3p | blv-miR-B3-5p | blv-miR-B4-3p | blv-miR-B4-5p | blv-miR-B5-3p | blv-miR-B5-5p | hsa-miR-106b-5p | hsa-miR-21-5p | hsa-mir-29a-3p |
|---|---|---|---|---|---|---|---|---|---|---|---|---|---|---|---|---|
| SRR25374679 | AML | 1.2M | 1.3M | 0.00 | 0.00 | 0.00 | 0.00 | 0.00 | 0.00 | 0.00 | 0.00 | 0.00 | 0.00 | 10.00 | 130.00 | 0.00 |
| SRR25374680 | AML | 0.8M | 0.8M | 0.00 | 0.00 | 0.00 | 0.00 | 0.00 | 0.00 | 0.00 | 1.00 | 0.00 | 0.00 | 0.00 | 50.00 | 0.00 |
| SRR25374681 | AML | 0.9M | 0.9M | 0.00 | 0.00 | 0.00 | 0.00 | 0.00 | 0.00 | 0.00 | 0.00 | 0.00 | 0.00 | 6.00 | 169.00 | 0.00 |
| SRR25374682 | AML | 1.1M | 1.1M | 0.00 | 0.00 | 0.00 | 0.00 | 0.00 | 0.00 | 0.00 | 0.00 | 0.00 | 0.00 | 5.00 | 342.00 | 0.00 |
| SRR25374683 | AML | 1.0M | 1.1M | 0.00 | 0.00 | 0.00 | 0.00 | 0.00 | 0.00 | 0.00 | 0.00 | 0.00 | 0.00 | 2.00 | 24.00 | 0.00 |
| SRR25374684 | AML | 0.7M | 0.7M | 0.00 | 0.00 | 0.00 | 0.00 | 0.00 | 0.00 | 0.00 | 0.00 | 0.00 | 0.00 | 5.00 | 32.00 | 0.00 |
| SRR25374685 | AML | 0.7M | 0.7M | 0.00 | 0.00 | 0.00 | 0.00 | 0.00 | 0.00 | 0.00 | 0.00 | 0.00 | 0.00 | 4.00 | 40.00 | 0.00 |
| SRR25374686 | AML | 0.9M | 0.9M | 0.00 | 0.00 | 0.00 | 0.00 | 0.00 | 0.00 | 0.00 | 0.00 | 0.00 | 0.00 | 7.00 | 106.00 | 2.00 |
| SRR25374687 | AML | 0.8M | 0.9M | 0.00 | 0.00 | 0.00 | 0.00 | 0.00 | 0.00 | 0.00 | 0.00 | 0.00 | 0.00 | 1.00 | 37.00 | 0.00 |
| SRR25374688 | AML | 0.8M | 0.9M | 0.00 | 0.00 | 0.00 | 0.00 | 0.00 | 0.00 | 0.00 | 0.00 | 0.00 | 0.00 | 4.00 | 27.00 | 0.00 |
| SRR25374689 | AML | 0.9M | 0.9M | 0.00 | 0.00 | 0.00 | 0.00 | 0.00 | 0.00 | 0.00 | 0.00 | 0.00 | 0.00 | 3.00 | 279.00 | 0.00 |
| SRR25374690 | AML | 1.1M | 1.2M | 0.00 | 1.00 | 0.00 | 0.00 | 0.00 | 0.00 | 0.00 | 0.00 | 0.00 | 0.00 | 9.00 | 108.00 | 0.00 |
| SRR25374691 | AML | 0.5M | 0.6M | 0.00 | 0.00 | 0.00 | 0.00 | 0.00 | 0.00 | 0.00 | 0.00 | 0.00 | 0.00 | 0.00 | 28.00 | 0.00 |
| SRR25374692 | AML | 1.3M | 1.3M | 0.00 | 0.00 | 0.00 | 0.00 | 0.00 | 0.00 | 0.00 | 0.00 | 0.00 | 0.00 | 11.00 | 141.00 | 0.00 |
| SRR25374693 | AML | 1.1M | 1.1M | 0.00 | 0.00 | 0.00 | 0.00 | 0.00 | 0.00 | 0.00 | 0.00 | 0.00 | 0.00 | 4.00 | 29.00 | 0.00 |
| SRR25374694 | AML | 1.6M | 1.6M | 0.00 | 0.00 | 0.00 | 0.00 | 0.00 | 0.00 | 0.00 | 0.00 | 0.00 | 0.00 | 26.00 | 117.00 | 0.00 |
| SRR25374695 | AML | 0.9M | 0.9M | 0.00 | 0.00 | 0.00 | 0.00 | 0.00 | 0.00 | 0.00 | 0.00 | 0.00 | 0.00 | 1.00 | 23.00 | 0.00 |
| SRR25374696 | AML | 1.4M | 1.4M | 0.00 | 0.00 | 0.00 | 0.00 | 0.00 | 0.00 | 0.00 | 0.00 | 0.00 | 0.00 | 5.00 | 31.00 | 0.00 |
| SRR25374697 | AML | 1.4M | 1.4M | 0.00 | 0.00 | 0.00 | 0.00 | 0.00 | 0.00 | 0.00 | 0.00 | 0.00 | 0.00 | 1.00 | 13.00 | 0.00 |
| SRR25374698 | AML | 1.0M | 1.0M | 0.00 | 0.00 | 0.00 | 0.00 | 0.00 | 0.00 | 0.00 | 0.00 | 0.00 | 0.00 | 2.00 | 117.00 | 0.00 |
| SRR25374699 | AML | 0.7M | 1.3M | 0.00 | 0.00 | 0.00 | 0.00 | 0.00 | 0.00 | 0.00 | 0.00 | 0.00 | 0.00 | 0.00 | 67.00 | 0.00 |
| SRR25374700 | AML | 1.0M | 1.2M | 0.00 | 0.00 | 0.00 | 0.00 | 0.00 | 0.00 | 0.00 | 0.00 | 0.00 | 0.00 | 1.00 | 17.00 | 0.00 |
| SRR25374701 | AML | 1.4M | 1.5M | 0.00 | 0.00 | 0.00 | 0.00 | 0.00 | 0.00 | 0.00 | 0.00 | 0.00 | 0.00 | 1.00 | 72.00 | 1.00 |
| SRR25374702 | AML | 1.4M | 1.4M | 0.00 | 0.00 | 0.00 | 0.00 | 0.00 | 0.00 | 0.00 | 0.00 | 0.00 | 0.00 | 5.00 | 223.00 | 1.00 |
| SRR25374703 | AML | 1.3M | 1.3M | 0.00 | 0.00 | 0.00 | 0.00 | 0.00 | 0.00 | 0.00 | 0.00 | 0.00 | 0.00 | 8.00 | 80.00 | 0.00 |
| SRR25374704 | AML | 1.4M | 1.5M | 0.00 | 0.00 | 0.00 | 0.00 | 0.00 | 0.00 | 0.00 | 0.00 | 0.00 | 0.00 | 0.00 | 493.00 | 0.00 |
| SRR25374705 | AML | 1.5M | 1.6M | 0.00 | 0.00 | 0.00 | 0.00 | 0.00 | 0.00 | 0.00 | 0.00 | 0.00 | 0.00 | 1.00 | 55.00 | 0.00 |
| SRR25374706 | AML | 1.7M | 1.7M | 0.00 | 0.00 | 0.00 | 0.00 | 0.00 | 0.00 | 0.00 | 0.00 | 0.00 | 0.00 | 0.00 | 50.00 | 0.00 |
| SRR25374707 | AML | 0.6M | 0.6M | 0.00 | 0.00 | 0.00 | 0.00 | 0.00 | 0.00 | 0.00 | 0.00 | 0.00 | 0.00 | 0.00 | 6.00 | 0.00 |
| SRR25374708 | AML | 1.2M | 1.2M | 0.00 | 0.00 | 0.00 | 0.00 | 0.00 | 0.00 | 0.00 | 0.00 | 0.00 | 0.00 | 2.00 | 241.00 | 0.00 |
| SRR25374709 | AML | 1.3M | 1.4M | 0.00 | 0.00 | 0.00 | 0.00 | 0.00 | 0.00 | 0.00 | 0.00 | 0.00 | 0.00 | 11.00 | 313.00 | 0.00 |
| SRR25374710 | AML | 1.5M | 1.6M | 0.00 | 0.00 | 0.00 | 0.00 | 0.00 | 0.00 | 0.00 | 0.00 | 0.00 | 0.00 | 6.00 | 285.00 | 0.00 |
| SRR25374711 | AML | 1.4M | 1.4M | 0.00 | 0.00 | 0.00 | 0.00 | 0.00 | 0.00 | 0.00 | 0.00 | 0.00 | 0.00 | 1.00 | 2.00 | 0.00 |
| SRR25374712 | AML | 1.2M | 1.3M | 0.00 | 0.00 | 0.00 | 0.00 | 0.00 | 0.00 | 0.00 | 0.00 | 0.00 | 0.00 | 2.00 | 17.00 | 0.00 |
| SRR25374713 | AML | 1.2M | 1.2M | 0.00 | 0.00 | 0.00 | 0.00 | 0.00 | 0.00 | 0.00 | 0.00 | 0.00 | 0.00 | 0.00 | 42.00 | 0.00 |
| SRR25374714 | AML | 1.4M | 1.6M | 0.00 | 0.00 | 0.00 | 0.00 | 0.00 | 0.00 | 0.00 | 0.00 | 0.00 | 0.00 | 0.00 | 131.00 | 0.00 |
| SRR25374715 | AML | 1.5M | 1.6M | 0.00 | 0.00 | 0.00 | 0.00 | 0.00 | 0.00 | 0.00 | 0.00 | 0.00 | 0.00 | 1.00 | 139.00 | 0.00 |
| SRR25374716 | AML | 0.9M | 1.0M | 0.00 | 0.00 | 0.00 | 0.00 | 0.00 | 0.00 | 0.00 | 0.00 | 0.00 | 0.00 | 0.00 | 26.00 | 0.00 |
| SRR25374717 | AML | 0.7M | 0.7M | 0.00 | 0.00 | 0.00 | 0.00 | 0.00 | 0.00 | 0.00 | 0.00 | 0.00 | 0.00 | 0.00 | 37.00 | 0.00 |
| SRR25374718 | AML | 0.4M | 0.4M | 0.00 | 0.00 | 0.00 | 0.00 | 0.00 | 0.00 | 0.00 | 0.00 | 0.00 | 0.00 | 0.00 | 15.00 | 0.00 |
| SRR25374719 | AML | 0.8M | 0.9M | 0.00 | 0.00 | 0.00 | 0.00 | 0.00 | 0.00 | 0.00 | 1.00 | 0.00 | 0.00 | 1.00 | 9.00 | 0.00 |
| SRR25374720 | AML | 0.7M | 0.7M | 0.00 | 0.00 | 0.00 | 0.00 | 0.00 | 0.00 | 0.00 | 0.00 | 0.00 | 0.00 | 1.00 | 24.00 | 1.00 |
| SRR25374721 | AML | 0.6M | 0.7M | 0.00 | 0.00 | 0.00 | 0.00 | 0.00 | 0.00 | 0.00 | 0.00 | 0.00 | 0.00 | 3.00 | 14.00 | 0.00 |
| SRR25374722 | AML | 0.9M | 0.9M | 0.00 | 0.00 | 0.00 | 0.00 | 0.00 | 0.00 | 0.00 | 0.00 | 0.00 | 0.00 | 0.00 | 68.00 | 0.00 |
| SRR25374723 | AML | 0.9M | 0.9M | 0.00 | 0.00 | 0.00 | 0.00 | 0.00 | 0.00 | 0.00 | 0.00 | 0.00 | 0.00 | 1.00 | 13.00 | 0.00 |
| SRR25374724 | AML | 0.9M | 0.9M | 0.00 | 0.00 | 0.00 | 0.00 | 0.00 | 0.00 | 0.00 | 0.00 | 0.00 | 0.00 | 0.00 | 11.00 | 0.00 |
| SRR25374725 | AML | 1.0M | 1.1M | 0.00 | 0.00 | 0.00 | 0.00 | 0.00 | 0.00 | 0.00 | 0.00 | 0.00 | 0.00 | 0.00 | 7.00 | 0.00 |
| SRR25374726 | AML | 0.4M | 0.4M | 0.00 | 0.00 | 0.00 | 0.00 | 0.00 | 0.00 | 0.00 | 0.00 | 0.00 | 0.00 | 3.00 | 27.00 | 0.00 |

(*Continued on next page*)

TABLE 2 PRJNA996975: Profiling of miRNA in pediatric AML in the Egyptian population[a] (Continued)

| Sample ID | Cancer type | Post trim reads | Total no of reads | blv-miR-B1-3p | blv-miR-B1-5p | blv-miR-B2-3p | blv-miR-B2-5p | blv-miR-B3-3p | blv-miR-B3-5p | blv-miR-B4-3p | blv-miR-B4-5p | blv-miR-B5-3p | blv-miR-B5-5p | hsa-miR-106b-5p | hsa-miR-21-5p | hsa-mir-29a-3p |
|---|---|---|---|---|---|---|---|---|---|---|---|---|---|---|---|---|
| | | | | | | | | | | | | | | **Number of reads that align on** | | |
| SRR25374727 | AML | 0.7M | 0.7M | 0.00 | 0.00 | 0.00 | 0.00 | 0.00 | 0.00 | 0.00 | 0.00 | 0.00 | 0.00 | 0.00 | 52.00 | 0.00 |
| SRR25374728 | NEGATIVE | 1.1M | 1.2M | 0.00 | 0.00 | 0.00 | 0.00 | 0.00 | 0.00 | 0.00 | 0.00 | 0.00 | 0.00 | 0.00 | 8.00 | 0.00 |
| SRR25374729 | AML | 0.9M | 0.9M | 0.00 | 0.00 | 0.00 | 0.00 | 0.00 | 0.00 | 0.00 | 0.00 | 0.00 | 0.00 | 1.00 | 413.00 | 0.00 |
| SRR25374730 | AML | 0.5M | 0.5M | 0.00 | 0.00 | 0.00 | 0.00 | 0.00 | 0.00 | 0.00 | 0.00 | 0.00 | 0.00 | 1.00 | 12.00 | 0.00 |
| SRR25374731 | AML | 0.7M | 0.7M | 0.00 | 0.00 | 0.00 | 0.00 | 0.00 | 0.00 | 0.00 | 0.00 | 0.00 | 0.00 | 1.00 | 228.00 | 0.00 |

[a]AML: acute myeloid leukemia.

to diverse industrial processes (i.e., pasteurization) (26–28). Moreover, the EVs containing miRNAs are stabilized in the gastrointestinal tract, favoring their absorption and subsequent transfer into the bloodstream of the individuals who ingest them (29–32). Furthermore, *in vitro* studies demonstrated (i) the ability of human cells to incorporate exogenous miRNAs from bovine vesicles (30, 33) and (ii) that BLV miRNAs can interact with human genes involved in tumor processes (19, 20). Given these observations, we sought to empirically determine whether BLV miRNAs could be detected in human cancers such as BCA, leukemia, and lymphoma. Rather than testing causation, we aimed to perform a high-sensitivity screen across multiple data sets to assess whether any detectable signal of BLV miRNAs exists in human cancer miRNomes.

## MATERIALS AND METHODS

### Human and animal data sets

We analyzed a total of 335 human miRNomes from nine different small RNAseq open studies from the NCBI (PRJNA758408, PRJNA792999, PRJNA934049, PRJNA1042957, PRJEB52224, PRJNA892942, PRJNA1024943, PRJNA996975, and PRJNA1126751) for the presence of BLV miRNAs. Two hundred twenty-eight samples from cancer patients: BCA, acute lymphoblastic leukemia (ALL), acute myeloid leukemia (AML), chronic lymphocytic leukemia (CLL), and 107 from healthy controls. To validate the detection sensitivity of our workflow, we also analyzed six bovine data sets (three BLV-infected and three uninfected; PRJNA378560) previously characterized for BLV miRNA expression.

TABLE 3 PRJNA1024943: Small RNA-seq reveals key miRNAs involved in cell differentiation arrest in AML[a,b]

| Sample ID | Cancer type | Post trim reads | Total no of reads | blv-miR-B1-3p | blv-miR-B1-5p | blv-miR-B2-3p | blv-miR-B2-5p | blv-miR-B3-3p | blv-miR-B3-5p | blv-miR-B4-3p | blv-miR-B4-5p | blv-miR-B5-3p | blv-miR-B5-5p | hsa-miR-106b-5p | hsa-miR-21-5p | hsa-mir-29a-3p |
|---|---|---|---|---|---|---|---|---|---|---|---|---|---|---|---|---|
| | | | | | | | | | | | | | | **Number of reads that align on** | | |
| SRR26304152 | AML CR | 9.1M | 9.9M | 0.00 | 0.00 | 0.00 | 0.00 | 0.00 | 0.00 | 0.00 | 1.00 | 0.00 | 0.00 | 3922.00 | 92,147.00 | 5473.00 |
| SRR26304153 | AML CR | 7.0M | 10.0M | 0.00 | 0.00 | 0.00 | 0.00 | 0.00 | 0.00 | 0.00 | 0.00 | 0.00 | 0.00 | 4064.00 | 171,965.00 | 41,921.00 |
| SRR26304154 | AML CR | 8.3M | 12.9M | 0.00 | 1.00 | 0.00 | 0.00 | 0.00 | 0.00 | 0.00 | 0.00 | 0.00 | 0.00 | 4769.00 | 302,613.00 | 40,566.00 |
| SRR26304155 | AML CR | 6.6M | 8.9M | 0.00 | 0.00 | 0.00 | 0.00 | 0.00 | 0.00 | 0.00 | 0.00 | 0.00 | 0.00 | 4930.00 | 226,499.00 | 45,406.00 |
| SRR26304156 | AML CR | 5.4M | 9.2M | 0.00 | 0.00 | 0.00 | 0.00 | 0.00 | 0.00 | 0.00 | 0.00 | 0.00 | 0.00 | 2683.00 | 165,299.00 | 22,155.00 |
| SRR26304157 | AML ID | 6.9M | 8.4M | 0.00 | 0.00 | 0.00 | 0.00 | 0.00 | 0.00 | 0.00 | 0.00 | 0.00 | 0.00 | 3986.00 | 124,081.00 | 15,695.00 |
| SRR26304158 | AML ID | 10.4M | 12.3M | 0.00 | 0.00 | 0.00 | 0.00 | 0.00 | 0.00 | 0.00 | 0.00 | 0.00 | 0.00 | 9171.00 | 144,842.00 | 30,344.00 |
| SRR26304159 | AML ID | 7.1M | 9.2M | 0.00 | 0.00 | 0.00 | 0.00 | 0.00 | 0.00 | 0.00 | 0.00 | 0.00 | 0.00 | 6178.00 | 69,138.00 | 13,496.00 |
| SRR26304160 | AML ID | 7.3M | 8.2M | 0.00 | 0.00 | 0.00 | 0.00 | 0.00 | 0.00 | 0.00 | 0.00 | 0.00 | 0.00 | 4175.00 | 271,174.00 | 37,096.00 |
| SRR26304161 | AML ID | 10.6M | 15.1M | 0.00 | 0.00 | 0.00 | 0.00 | 0.00 | 0.00 | 0.00 | 0.00 | 0.00 | 0.00 | 8840.00 | 201,785.00 | 42,485.00 |

[a]AML CR: acute myeloid leukemia—complete remission.
[b]AML ID: acute myeloid leukemia—initial diagnosis.

**TABLE 4**  PRJNA892942: HDAC1 regulates the chromatin landscape to control transcriptional dependencies in CLL[a]

| Sample ID | Cancer type | Post trim reads | Total no of reads | blv-miR-B1-3p | blv-miR-B1-5p | blv-miR-B2-3p | blv-miR-B2-5p | blv-miR-B3-3p | blv-miR-B3-5p | blv-miR-B4-3p | blv-miR-B4-5p | blv-miR-B5-3p | blv-miR-B5-5p | hsa-miR-106b-5p | hsa-miR-21-5p | hsa-mir-29a-3p |
|---|---|---|---|---|---|---|---|---|---|---|---|---|---|---|---|---|
| | | | | | | | | | | | | | | Number of reads that align on | | |
| SRR22000574 | CLL | 11.4M | 11.4M | 0.00 | 0.00 | 0.00 | 0.00 | 0.00 | 0.00 | 0.00 | 0.00 | 0.00 | 0.00 | 2128.00 | 1,223,652.00 | 206,950.00 |
| SRR22000575 | CLL | 11.0M | 11.0M | 0.00 | 0.00 | 0.00 | 0.00 | 0.00 | 0.00 | 0.00 | 0.00 | 0.00 | 0.00 | 1666.00 | 451,575.00 | 129,685.00 |
| SRR22000576 | CLL | 10.8M | 10.8M | 0.00 | 0.00 | 0.00 | 0.00 | 0.00 | 0.00 | 0.00 | 0.00 | 0.00 | 0.00 | 1418.00 | 1,080,412.00 | 148,536.00 |
| SRR22000577 | CLL | 14.5M | 14.5M | 0.00 | 0.00 | 0.00 | 0.00 | 0.00 | 0.00 | 0.00 | 0.00 | 0.00 | 0.00 | 2676.00 | 616,703.00 | 211,004.00 |
| SRR22000578 | CLL | 17.7M | 17.7M | 0.00 | 0.00 | 0.00 | 0.00 | 0.00 | 0.00 | 0.00 | 0.00 | 0.00 | 0.00 | 2685.00 | 1,450,078.00 | 139,692.00 |
| SRR22000579 | CLL | 20.5M | 20.5M | 0.00 | 0.00 | 0.00 | 0.00 | 0.00 | 0.00 | 0.00 | 0.00 | 0.00 | 0.00 | 3777.00 | 711,561.00 | 161,183.00 |
| SRR22000580 | CLL | 8.5M | 8.5M | 0.00 | 0.00 | 0.00 | 0.00 | 0.00 | 0.00 | 0.00 | 0.00 | 0.00 | 0.00 | 1608.00 | 346,386.00 | 65,391.00 |
| SRR22000581 | CLL | 9.5M | 9.5M | 0.00 | 0.00 | 0.00 | 0.00 | 0.00 | 0.00 | 0.00 | 0.00 | 0.00 | 0.00 | 1182.00 | 242,849.00 | 40,059.00 |
| SRR22000582 | CLL | 7.6M | 7.6M | 0.00 | 0.00 | 0.00 | 0.00 | 0.00 | 0.00 | 0.00 | 0.00 | 0.00 | 0.00 | 935.00 | 314,786.00 | 58,362.00 |
| SRR22000583 | CLL | 4.2M | 4.2M | 0.00 | 0.00 | 0.00 | 0.00 | 0.00 | 0.00 | 0.00 | 0.00 | 0.00 | 0.00 | 839.00 | 128,651.00 | 59,216.00 |
| SRR22000584 | CLL | 12.1M | 12.1M | 0.00 | 0.00 | 0.00 | 0.00 | 0.00 | 0.00 | 0.00 | 0.00 | 0.00 | 0.00 | 1133.00 | 487,541.00 | 62,189.00 |
| SRR22000585 | CLL | 12.4M | 12.4M | 0.00 | 0.00 | 0.00 | 0.00 | 0.00 | 0.00 | 0.00 | 0.00 | 0.00 | 0.00 | 2011.00 | 403,692.00 | 81,572.00 |
| SRR22000586 | CLL | 14.5M | 14.5M | 0.00 | 0.00 | 0.00 | 0.00 | 0.00 | 0.00 | 0.00 | 0.00 | 0.00 | 0.00 | 2279.00 | 471,368.00 | 206,998.00 |
| SRR22000587 | CLL | 18.7M | 18.7M | 0.00 | 0.00 | 0.00 | 0.00 | 0.00 | 0.00 | 0.00 | 0.00 | 0.00 | 0.00 | 2199.00 | 362,456.00 | 178,743.00 |
| SRR22000588 | CLL | 6.7M | 6.7M | 0.00 | 0.00 | 0.00 | 0.00 | 0.00 | 0.00 | 0.00 | 0.00 | 0.00 | 0.00 | 1302.00 | 513,670.00 | 81,056.00 |
| SRR22000589 | CLL | 8.5M | 8.5M | 0.00 | 0.00 | 0.00 | 0.00 | 0.00 | 0.00 | 0.00 | 0.00 | 0.00 | 0.00 | 777.00 | 225,901.00 | 54,010.00 |
| SRR22000590 | CLL | 11.4M | 11.4M | 0.00 | 0.00 | 0.00 | 0.00 | 0.00 | 0.00 | 0.00 | 0.00 | 0.00 | 0.00 | 2136.00 | 520,223.00 | 210,896.00 |
| SRR22000591 | CLL | 14.2M | 14.2M | 0.00 | 2.00 | 0.00 | 0.00 | 0.00 | 0.00 | 0.00 | 0.00 | 0.00 | 0.00 | 1748.00 | 414,372.00 | 155,670.00 |
| SRR22000592 | CLL | 9.1M | 9.1M | 0.00 | 0.00 | 0.00 | 0.00 | 0.00 | 0.00 | 0.00 | 0.00 | 0.00 | 0.00 | 1978.00 | 428,631.00 | 99,983.00 |
| SRR22000593 | CLL | 13.4M | 13.4M | 0.00 | 0.00 | 0.00 | 0.00 | 0.00 | 0.00 | 0.00 | 0.00 | 0.00 | 0.00 | 2986.00 | 240,818.00 | 145,067.00 |

[a]CLL: chronic lymphocytic leukemia.

## Small RNA sequencing reads processing

All data analyses were performed using the Galaxy platform (version 24.1.4) (https://usegalaxy.org/). Raw fastq files were transferred from NCBI (https://www.ncbi.nlm.nih.gov/sra) to the Galaxy server for downstream analysis. Quality evaluation of the raw sequence data was done with FastQC (Version 0.12.1) (http://www.bioinformatics.babraham.ac.uk/projects/fastqc/). The basic statistics, the sequence quality, quality score, sequence content, and GC content of each raw sequence were evaluated, and raw data that fulfilled basic quality parameters were used for detection analysis. Adapters (i.e., Illumina Universal and Illumina Small Seq) and low-quality ends were trimmed using CutAdapt (version 4.9) (34) and/ or Trim Galore (version 0.6.7) (https://www.bioinformatics.babraham.ac.uk/projects/trim_galore/), and sequences with less than 18 bp or more than 30 bp were removed from downstream analysis. Following this, the fastq files were parsed into fasta format, sequences with non-canonical nucleotides were discarded, and identical reads were collapsed using the miRDeep2 mapper tool (version 2.0.0.8) (35).

## miRNAs identification and quantification

Prior to sequence alignment, all BLV and control miRNAs and hairpin (precursor) reference sequences (Table S1) were downloaded from the miRbase (https://www.mir-base.org/) (36) and uploaded to Galaxy.org in fasta format. We searched for BLV miRNAs in human cancer samples using miRDeep2 quantifier. We deliberately used this tool for the detection and quantification of known BLV miRNAs with maximal sensitivity. We used collapsed sRNA-seq reads and mature and precursor BLV-encoded miRNAs as inputs. miRNA quantification involved first mapping reads to miRNA precursor sequences, followed by mapping to corresponding mature miRNAs. We allowed up to two mismatches to increase sensitivity. Normalization was omitted to avoid losing

**TABLE 5** PRJEB52224: miRNA-sequencing of Brazilian pediatric ALL[a]

| Sample ID | Cancer type | Post trim reads | Total no of reads | blv-miR-B1-3p | blv-miR-B1-5p | blv-miR-B2-3p | blv-miR-B2-5p | blv-miR-B3-3p | blv-miR-B3-5p | blv-miR-B4-3p | blv-miR-B4-5p | blv-miR-B5-3p | blv-miR-B5-5p | hsa-miR-106b-5p | hsa-miR-21-5p | hsa-mir-29a-3p |
|---|---|---|---|---|---|---|---|---|---|---|---|---|---|---|---|---|
| ERR9566615 | B-cell ALL | 7.4M | 8.7M | 0.00 | 0.00 | 0.00 | 0.00 | 0.00 | 0.00 | 0.00 | 0.00 | 0.00 | 1.00 | 4249.00 | 29,214.00 | 3955.00 |
| ERR9566616 | B-cell ALL | 7.4M | 8.8M | 0.00 | 0.00 | 0.00 | 0.00 | 0.00 | 0.00 | 0.00 | 0.00 | 0.00 | 0.00 | 4305.00 | 29,472.00 | 4078.00 |
| ERR9566617 | B-cell ALL | 6.8M | 8.2M | 0.00 | 0.00 | 0.00 | 0.00 | 0.00 | 0.00 | 0.00 | 0.00 | 0.00 | 0.00 | 2752.00 | 31,394.00 | 4517.00 |
| ERR9566618 | B-cell ALL | 6.8M | 8.2M | 0.00 | 0.00 | 0.00 | 0.00 | 0.00 | 0.00 | 1.00 | 0.00 | 0.00 | 0.00 | 2783.00 | 31,711.00 | 4575.00 |
| ERR9566619 | T-cell ALL | 0.7M | 0.8M | 0.00 | 0.00 | 0.00 | 0.00 | 0.00 | 0.00 | 0.00 | 0.00 | 0.00 | 0.00 | 35.00 | 1288.00 | 118.00 |
| ERR9566620 | T-cell ALL | 0.7M | 0.8M | 0.00 | 0.00 | 0.00 | 0.00 | 0.00 | 0.00 | 0.00 | 0.00 | 0.00 | 0.00 | 35.00 | 1295.00 | 94.00 |
| ERR9566621 | B-cell ALL | 8.6M | 9.2M | 0.00 | 0.00 | 0.00 | 0.00 | 0.00 | 0.00 | 0.00 | 0.00 | 0.00 | 0.00 | 8594.00 | 29,879.00 | 4650.00 |
| ERR9566622 | B-cell ALL | 8.6M | 9.2M | 0.00 | 0.00 | 0.00 | 0.00 | 0.00 | 0.00 | 0.00 | 0.00 | 0.00 | 0.00 | 8262.00 | 30,113.00 | 4712.00 |
| ERR9566623 | B-cell ALL | 5.8M | 6.4M | 0.00 | 0.00 | 0.00 | 0.00 | 0.00 | 0.00 | 0.00 | 0.00 | 0.00 | 0.00 | 2529.00 | 40,542.00 | 2914.00 |
| ERR9566624 | B-cell ALL | 5.8M | 6.3M | 0.00 | 0.00 | 0.00 | 0.00 | 0.00 | 0.00 | 0.00 | 0.00 | 0.00 | 0.00 | 2578.00 | 39,850.00 | 2891.00 |
| ERR9566625 | B-cell ALL | 6.8M | 7.4M | 0.00 | 0.00 | 0.00 | 0.00 | 0.00 | 0.00 | 0.00 | 0.00 | 0.00 | 0.00 | 1294.00 | 27,291.00 | 5522.00 |
| ERR9566626 | B-cell ALL | 6.9M | 7.5M | 0.00 | 0.00 | 0.00 | 0.00 | 0.00 | 0.00 | 0.00 | 0.00 | 0.00 | 0.00 | 1342.00 | 27,457.00 | 5701.00 |
| ERR9566627 | B-cell ALL | 9.5M | 10.3M | 0.00 | 0.00 | 0.00 | 0.00 | 0.00 | 0.00 | 0.00 | 0.00 | 0.00 | 0.00 | 7039.00 | 31,058.00 | 5159.00 |
| ERR9566628 | B-cell ALL | 9.5M | 10.4M | 0.00 | 0.00 | 0.00 | 0.00 | 0.00 | 0.00 | 0.00 | 0.00 | 0.00 | 0.00 | 6850.00 | 31,330.00 | 5247.00 |
| ERR9566629 | T-cell ALL | 4.7M | 5.8M | 0.00 | 0.00 | 0.00 | 0.00 | 0.00 | 0.00 | 0.00 | 0.00 | 0.00 | 0.00 | 2487.00 | 23,154.00 | 2433.00 |
| ERR9566630 | T-cell ALL | 4.7M | 5.8M | 0.00 | 0.00 | 0.00 | 0.00 | 0.00 | 0.00 | 0.00 | 0.00 | 0.00 | 0.00 | 2507.00 | 22,762.00 | 2318.00 |
| ERR9566631 | T-cell ALL | 4.8M | 5.2M | 0.00 | 0.00 | 0.00 | 0.00 | 0.00 | 0.00 | 0.00 | 0.00 | 0.00 | 0.00 | 2837.00 | 17,828.00 | 2853.00 |
| ERR9566632 | T-cell ALL | 4.8M | 5.2M | 0.00 | 0.00 | 0.00 | 0.00 | 0.00 | 0.00 | 0.00 | 0.00 | 0.00 | 0.00 | 2669.00 | 17,277.00 | 2847.00 |
| ERR9566633 | T-cell ALL | 3.1M | 3.4M | 0.00 | 0.00 | 0.00 | 0.00 | 0.00 | 0.00 | 0.00 | 0.00 | 0.00 | 0.00 | 1486.00 | 10,149.00 | 2155.00 |
| ERR9566634 | T-cell ALL | 3.1M | 3.5M | 0.00 | 0.00 | 0.00 | 0.00 | 0.00 | 0.00 | 0.00 | 0.00 | 0.00 | 0.00 | 1416.00 | 10,213.00 | 2275.00 |
| ERR9566635 | T-cell ALL | 7.3M | 9.9M | 0.00 | 0.00 | 0.00 | 0.00 | 0.00 | 0.00 | 0.00 | 0.00 | 0.00 | 0.00 | 3681.00 | 56,502.00 | 7791.00 |
| ERR9566636 | T-cell ALL | 7.3M | 9.8M | 0.00 | 0.00 | 0.00 | 0.00 | 0.00 | 0.00 | 0.00 | 0.00 | 0.00 | 0.00 | 3709.00 | 55,543.00 | 7730.00 |
| ERR9566637 | B-cell ALL | 5.3M | 5.7M | 0.00 | 0.00 | 0.00 | 0.00 | 0.00 | 0.00 | 0.00 | 0.00 | 0.00 | 0.00 | 1594.00 | 7260.00 | 1853.00 |
| ERR9566638 | B-cell ALL | 5.4M | 5.7M | 0.00 | 0.00 | 0.00 | 0.00 | 0.00 | 0.00 | 0.00 | 0.00 | 0.00 | 0.00 | 1520.00 | 7340.00 | 1890.00 |
| ERR9566639 | T-cell ALL | 4.8M | 7.2M | 0.00 | 0.00 | 0.00 | 0.00 | 0.00 | 0.00 | 0.00 | 0.00 | 0.00 | 0.00 | 1209.00 | 15,926.00 | 1057.00 |
| ERR9566640 | T-cell ALL | 4.8M | 7.3M | 0.00 | 0.00 | 0.00 | 0.00 | 0.00 | 0.00 | 1.00 | 0.00 | 0.00 | 0.00 | 1222.00 | 16,226.00 | 1117.00 |
| ERR9566641 | T-cell ALL | 5.9M | 6.7M | 0.00 | 0.00 | 0.00 | 0.00 | 0.00 | 0.00 | 0.00 | 0.00 | 0.00 | 0.00 | 1807.00 | 79,719.00 | 2335.00 |
| ERR9566642 | T-cell ALL | 6.0M | 6.7M | 0.00 | 0.00 | 0.00 | 0.00 | 0.00 | 0.00 | 0.00 | 0.00 | 0.00 | 0.00 | 1738.00 | 81,115.00 | 2366.00 |
| ERR9566643 | B-cell ALL | 4.1M | 4.5M | 0.00 | 0.00 | 0.00 | 0.00 | 0.00 | 0.00 | 0.00 | 0.00 | 0.00 | 0.00 | 586.00 | 8484.00 | 1277.00 |
| ERR9566644 | B-cell ALL | 4.1M | 4.5M | 0.00 | 0.00 | 0.00 | 0.00 | 0.00 | 0.00 | 0.00 | 0.00 | 0.00 | 0.00 | 606.00 | 8367.00 | 1297.00 |

[a]ALL: acute lymphoblastic leukemia.

weak signals, as the study's aim was not to assess differential expression but rather the presence of BLV miRNAs. Additionally, a set of human miRNAs (hsa-mir-29a-3p, hsa-miR-106b-5p, and hsa-miR-21-5p) served as a positive control for data integrity. CSV files generated by the quantifier module of the miRDeep2 software were downloaded, modified, and shown in Tables 1 to 9.

## RESULTS

We analyzed a total of 335 human samples from nine different small RNAseq studies using the Galaxy.org platform. A total of 228 cancer samples: BCA, ALL, AML, CLL, and 107 controls were tested for the presence of BLV-miRNAs. A total of 1.98 billion trimmed reads (3.1 billion raw) from human cancer and control data sets were analyzed. Only 60 reads (0.000003%) aligned to BLV miRNA reference sequences, most mapping to STAR strands and/or containing up to two mismatches (Tables 1 to 9). For example, SRR15658028 (PRJNA758408), which had 11 reads mapping to blv-miR-B4-5p STAR strands, had two substitutions compared to the reference sequence (Fig. S1). The detection of human miRNA controls confirmed data quality across most data sets, except

TABLE 6 PRJNA1042957: The miRNA expression of 5 normal breast tissues and 15 BCA tissues[a]

| Sample ID | Cancer type | Post trim reads | Total no of reads | Number of reads that align on | | | | | | | | | | | | |
|---|---|---|---|---|---|---|---|---|---|---|---|---|---|---|---|---|
| | | | | blv-miR-B1-3p | blv-miR-B1-5p | blv-miR-B2-3p | blv-miR-B2-5p | blv-miR-B3-3p | blv-miR-B3-5p | blv-miR-B4-3p | blv-miR-B4-5p | blv-miR-B5-3p | blv-miR-B5-5p | hsa-miR-106b-5p | hsa-miR-21-5p | hsa-mir-29a-3p |
| SRR26902945 | BCA | 9.3M | 11.9M | 0.00 | 0.00 | 0.00 | 0.00 | 0.00 | 0.00 | 1.00 | 0.00 | 0.00 | 0.00 | 1865.00 | 412,296.00 | 17,420.00 |
| SRR26902946 | BCA | 9.1M | 11.6M | 0.00 | 0.00 | 0.00 | 0.00 | 0.00 | 0.00 | 0.00 | 0.00 | 0.00 | 0.00 | 2582.00 | 545,772.00 | 20,804.00 |
| SRR26902947 | BCA | 10.9M | 12.7M | 0.00 | 0.00 | 0.00 | 0.00 | 0.00 | 0.00 | 0.00 | 0.00 | 0.00 | 0.00 | 1972.00 | 1,199,853.00 | 16,764.00 |
| SRR26902948 | BCA | 10.3M | 12.0M | 0.00 | 0.00 | 0.00 | 0.00 | 0.00 | 0.00 | 0.00 | 0.00 | 0.00 | 0.00 | 1372.00 | 729,057.00 | 10,757.00 |
| SRR26902949 | BCA | 9.1M | 12.6M | 0.00 | 0.00 | 0.00 | 0.00 | 0.00 | 0.00 | 0.00 | 0.00 | 0.00 | 1.00 | 1040.00 | 692,073.00 | 17,156.00 |
| SRR26902950 | NEGATIVE | 11.3M | 13.0M | 0.00 | 0.00 | 0.00 | 0.00 | 0.00 | 0.00 | 0.00 | 0.00 | 0.00 | 0.00 | 1545.00 | 128,465.00 | 29,956.00 |
| SRR26902951 | NEGATIVE | 12.4M | 13.8M | 0.00 | 0.00 | 0.00 | 0.00 | 0.00 | 0.00 | 0.00 | 0.00 | 0.00 | 0.00 | 2254.00 | 111,565.00 | 16,544.00 |
| SRR26902952 | NEGATIVE | 11.7M | 12.9M | 0.00 | 0.00 | 0.00 | 0.00 | 0.00 | 0.00 | 0.00 | 0.00 | 0.00 | 0.00 | 1072.00 | 84,363.00 | 29,109.00 |
| SRR26902953 | BCA | 9.8M | 14.0M | 0.00 | 0.00 | 0.00 | 0.00 | 0.00 | 0.00 | 0.00 | 0.00 | 0.00 | 1.00 | 3683.00 | 584,070.00 | 6014.00 |
| SRR26902954 | BCA | 10.4M | 14.1M | 0.00 | 0.00 | 0.00 | 0.00 | 0.00 | 0.00 | 0.00 | 0.00 | 0.00 | 0.00 | 7518.00 | 877,847.00 | 48,694.00 |
| SRR26902955 | BCA | 12.3M | 13.8M | 0.00 | 0.00 | 0.00 | 0.00 | 0.00 | 0.00 | 0.00 | 0.00 | 0.00 | 1.00 | 5621.00 | 1,325,291.00 | 37,203.00 |
| SRR26902956 | BCA | 12.0M | 13.1M | 0.00 | 0.00 | 0.00 | 0.00 | 0.00 | 0.00 | 0.00 | 0.00 | 0.00 | 0.00 | 1869.00 | 685,377.00 | 36,759.00 |
| SRR26902957 | BCA | 14.1M | 15.9M | 0.00 | 0.00 | 0.00 | 0.00 | 0.00 | 0.00 | 0.00 | 0.00 | 0.00 | 0.00 | 9390.00 | 650,746.00 | 24,473.00 |
| SRR26902958 | BCA | 12.1M | 16.1M | 0.00 | 1.00 | 0.00 | 0.00 | 0.00 | 0.00 | 0.00 | 0.00 | 0.00 | 0.00 | 3163.00 | 641,472.00 | 47,566.00 |
| SRR26902959 | BCA | 13.1M | 14.9M | 0.00 | 1.00 | 0.00 | 0.00 | 0.00 | 0.00 | 0.00 | 0.00 | 0.00 | 0.00 | 5900.00 | 944,153.00 | 19,620.00 |
| SRR26902960 | BCA | 14.7M | 15.9M | 0.00 | 0.00 | 0.00 | 0.00 | 0.00 | 0.00 | 0.00 | 0.00 | 0.00 | 0.00 | 4150.00 | 1,614,219.00 | 35,064.00 |
| SRR26902961 | BCA | 10.6M | 13.2M | 0.00 | 0.00 | 0.00 | 0.00 | 0.00 | 0.00 | 0.00 | 0.00 | 0.00 | 0.00 | 19,870.00 | 932,350.00 | 14,936.00 |
| SRR26902962 | BCA | 11.2M | 15.1M | 0.00 | 0.00 | 0.00 | 0.00 | 0.00 | 0.00 | 0.00 | 0.00 | 0.00 | 0.00 | 6504.00 | 1,201,883.00 | 14,219.00 |
| SRR26902963 | NEGATIVE | 11.2M | 14.3M | 0.00 | 0.00 | 0.00 | 0.00 | 0.00 | 0.00 | 0.00 | 0.00 | 0.00 | 0.00 | 209.00 | 26,355.00 | 6327.00 |
| SRR26902964 | NEGATIVE | 12.8M | 13.5M | 0.00 | 0.00 | 0.00 | 0.00 | 0.00 | 0.00 | 0.00 | 0.00 | 0.00 | 0.00 | 1409.00 | 122,413.00 | 36,674.00 |

[a]BCA: breast cancer.

for PRJNA996975, which was excluded due to low sequencing depth and poor recovery of internal positive controls.

Additionally, a subset of three small RNA-seq data sets from white blood cells of cattle infected and three from cattle non-infected with BLV were analyzed to validate the analytical workflow in biological samples. In cattle data sets, 9 of 10 BLV miRNAs were robustly detected in BLV-positive samples, validating our workflow (Table 10). BLV miRNAs were detected at lower frequencies in BLV-uninfected cattle and were attributed to background noise or early infection stages. Normalized abundance (RPM) confirmed a >200-fold difference between infected and uninfected cattle (10669 vs. 42 RPM). These results demonstrate that the pipeline can reliably detect BLV miRNAs when present in biological samples.

## DISCUSSION

Several viruses are known to contribute to the development of human malignancies (37); however, it is not clear whether BLV exposure might augment the risk of BCA in humans (2, 38). While some studies support this hypothesis (3–13), other studies argue against it (14–16). In previous studies, whole-genome and whole-transcriptome sequencing analysis of human BCA samples failed to detect sequencing reads corresponding to BLV proviral DNA or viral transcripts in these samples (15, 16). This study explored an alternative viral factor potentially involved in oncogenesis. BLV encodes a set of miRNAs that are highly expressed during natural BLV infection in cattle (22, 39), and some of these miRNAs could interact with human tumoral suppressor genes (19, 20). Interestingly, bovine miRNAs can be conveyed through EVs present in milk from cattle (25, 33, 40). Moreover, EVs containing miRNAs are absorbed in the intestine and subsequently transferred into the bloodstream of the individuals who ingest them (29–31). Different human cells (i.e., hepatic cells, macrophages, PBMCs) can incorporate

**TABLE 7** PRJNA934049: Circulating miRNA biomarker for detecting BCA in high-risk benign breast tumors[a,b,c]

| Sample ID | Cancer type | Post trim reads | Total no of reads | Number of reads that align on | | | | | | | | | | | | |
|---|---|---|---|---|---|---|---|---|---|---|---|---|---|---|---|---|
| | | | | blv-miR-B1-3p | blv-miR-B1-5p | blv-miR-B2-3p | blv-miR-B2-5p | blv-miR-B3-3p | blv-miR-B3-5p | blv-miR-B4-3p | blv-miR-B4-5p | blv-miR-B5-3p | blv-miR-B5-5p | hsa-miR-106b-5p | hsa-miR-21-5p | hsa-mir-29a-3p |
| SRR23425832 | HRB | 6.0M | 9.3M | 0.00 | 0.00 | 0.00 | 0.00 | 0.00 | 0.00 | 0.00 | 0.00 | 0.00 | 0.00 | 57.00 | 28,461.00 | 3807.00 |
| SRR23425833 | HRB | 6.9M | 9.4M | 0.00 | 0.00 | 0.00 | 0.00 | 0.00 | 0.00 | 0.00 | 0.00 | 0.00 | 0.00 | 148.00 | 16,671.00 | 4156.00 |
| SRR23425834 | HRB | 4.4M | 7.5M | 0.00 | 0.00 | 0.00 | 0.00 | 0.00 | 0.00 | 0.00 | 0.00 | 0.00 | 0.00 | 71.00 | 9938.00 | 1852.00 |
| SRR23425835 | HRB | 5.4M | 11.6M | 0.00 | 0.00 | 0.00 | 0.00 | 0.00 | 0.00 | 0.00 | 0.00 | 0.00 | 0.00 | 33.00 | 9327.00 | 1320.00 |
| SRR23425836 | HRB | 5.9M | 8.8M | 0.00 | 0.00 | 0.00 | 0.00 | 0.00 | 0.00 | 0.00 | 0.00 | 0.00 | 0.00 | 65.00 | 17,191.00 | 2073.00 |
| SRR23425837 | HRB | 6.7M | 11.3M | 0.00 | 0.00 | 0.00 | 0.00 | 0.00 | 0.00 | 0.00 | 0.00 | 0.00 | 0.00 | 68.00 | 25,093.00 | 2631.00 |
| SRR23425838 | HRB | 6.1M | 9.2M | 0.00 | 0.00 | 0.00 | 0.00 | 1.00 | 0.00 | 0.00 | 0.00 | 0.00 | 0.00 | 66.00 | 24,666.00 | 2331.00 |
| SRR23425839 | HRB | 6.5M | 10.2M | 0.00 | 0.00 | 0.00 | 0.00 | 0.00 | 0.00 | 0.00 | 0.00 | 0.00 | 1.00 | 62.00 | 24,237.00 | 2600.00 |
| SRR23425840 | HRB | 7.8M | 12.6M | 0.00 | 0.00 | 0.00 | 0.00 | 0.00 | 0.00 | 0.00 | 0.00 | 0.00 | 0.00 | 67.00 | 31,239.00 | 3107.00 |
| SRR23425841 | HRB | 4.9M | 9.8M | 0.00 | 0.00 | 0.00 | 0.00 | 0.00 | 0.00 | 0.00 | 0.00 | 0.00 | 0.00 | 57.00 | 14,946.00 | 2709.00 |
| SRR23425842 | HRB | 52.8M | 92.6M | 0.00 | 0.00 | 0.00 | 0.00 | 0.00 | 0.00 | 0.00 | 0.00 | 0.00 | 0.00 | 455.00 | 150,618.00 | 30,103.00 |
| SRR23425843 | HRB | 5.6M | 10.3M | 0.00 | 0.00 | 0.00 | 0.00 | 0.00 | 0.00 | 0.00 | 0.00 | 0.00 | 0.00 | 54.00 | 19,832.00 | 3748.00 |
| SRR23425844 | HRB | 6.3M | 11.2M | 0.00 | 0.00 | 0.00 | 0.00 | 0.00 | 0.00 | 0.00 | 0.00 | 0.00 | 0.00 | 126.00 | 46,332.00 | 9883.00 |
| SRR23425845 | HRB | 6.0M | 8.9M | 0.00 | 0.00 | 0.00 | 0.00 | 0.00 | 0.00 | 0.00 | 0.00 | 0.00 | 0.00 | 25.00 | 21,386.00 | 1679.00 |
| SRR23425846 | HRB | 5.8M | 8.1M | 0.00 | 0.00 | 0.00 | 0.00 | 0.00 | 0.00 | 0.00 | 0.00 | 0.00 | 3.00 | 26.00 | 8207.00 | 1701.00 |
| SRR23425847 | HRB | 7.5M | 8.9M | 0.00 | 0.00 | 0.00 | 0.00 | 0.00 | 0.00 | 0.00 | 0.00 | 0.00 | 0.00 | 72.00 | 96,872.00 | 5920.00 |
| SRR23425848 | HRB | 7.7M | 9.9M | 0.00 | 0.00 | 0.00 | 0.00 | 0.00 | 0.00 | 0.00 | 0.00 | 0.00 | 0.00 | 74.00 | 31,431.00 | 3323.00 |
| SRR23425849 | HRB | 4.9M | 9.1M | 0.00 | 0.00 | 0.00 | 0.00 | 0.00 | 0.00 | 0.00 | 0.00 | 0.00 | 0.00 | 28.00 | 33,311.00 | 2283.00 |
| SRR23425850 | ESC | 5.0M | 8.9M | 0.00 | 0.00 | 0.00 | 0.00 | 0.00 | 0.00 | 0.00 | 0.00 | 0.00 | 0.00 | 39.00 | 34,418.00 | 1967.00 |
| SRR23425851 | ESC | 3.5M | 8.7M | 0.00 | 0.00 | 0.00 | 0.00 | 0.00 | 0.00 | 0.00 | 0.00 | 0.00 | 0.00 | 4.00 | 12,939.00 | 1223.00 |
| SRR23425852 | ESC | 5.4M | 8.6M | 0.00 | 0.00 | 0.00 | 0.00 | 0.00 | 0.00 | 0.00 | 0.00 | 0.00 | 0.00 | 71.00 | 31,484.00 | 4186.00 |
| SRR23425853 | ESC | 2.4M | 8.5M | 0.00 | 0.00 | 0.00 | 0.00 | 0.00 | 0.00 | 0.00 | 0.00 | 0.00 | 0.00 | 14.00 | 10,472.00 | 1332.00 |
| SRR23425854 | ESC | 4.7M | 7.8M | 0.00 | 0.00 | 0.00 | 0.00 | 0.00 | 0.00 | 0.00 | 0.00 | 0.00 | 0.00 | 20.00 | 16,228.00 | 2676.00 |
| SRR23425855 | ESC | 6.8M | 7.8M | 0.00 | 0.00 | 0.00 | 0.00 | 0.00 | 0.00 | 0.00 | 0.00 | 0.00 | 0.00 | 109.00 | 72,177.00 | 3692.00 |
| SRR23425856 | ESC | 5.1M | 8.6M | 0.00 | 0.00 | 0.00 | 0.00 | 0.00 | 0.00 | 0.00 | 0.00 | 0.00 | 1.00 | 69.00 | 21,932.00 | 2039.00 |
| SRR23425857 | ESC | 4.6M | 8.6M | 0.00 | 0.00 | 0.00 | 0.00 | 0.00 | 0.00 | 0.00 | 0.00 | 0.00 | 0.00 | 46.00 | 19,768.00 | 1985.00 |
| SRR23425858 | ESC | 4.6M | 7.4M | 0.00 | 0.00 | 0.00 | 0.00 | 0.00 | 0.00 | 0.00 | 0.00 | 0.00 | 0.00 | 49.00 | 22,279.00 | 2133.00 |
| SRR23425859 | NRB | 4.1M | 7.2M | 0.00 | 0.00 | 0.00 | 0.00 | 0.00 | 0.00 | 0.00 | 0.00 | 0.00 | 2.00 | 48.00 | 16,368.00 | 1670.00 |
| SRR23425860 | NRB | 5.3M | 8.9M | 0.00 | 0.00 | 0.00 | 0.00 | 0.00 | 0.00 | 0.00 | 0.00 | 0.00 | 0.00 | 62.00 | 21,002.00 | 2427.00 |
| SRR23425861 | NRB | 5.5M | 8.3M | 0.00 | 0.00 | 0.00 | 0.00 | 0.00 | 0.00 | 0.00 | 0.00 | 0.00 | 0.00 | 62.00 | 26,416.00 | 2841.00 |
| SRR23425862 | NRB | 4.5M | 7.5M | 0.00 | 0.00 | 0.00 | 0.00 | 0.00 | 0.00 | 0.00 | 0.00 | 0.00 | 0.00 | 58.00 | 18,479.00 | 1847.00 |
| SRR23425863 | NRB | 6.3M | 10.8M | 0.00 | 0.00 | 0.00 | 0.00 | 0.00 | 0.00 | 0.00 | 0.00 | 0.00 | 1.00 | 60.00 | 25,429.00 | 2642.00 |
| SRR23425864 | NRB | 4.0M | 8.3M | 0.00 | 0.00 | 0.00 | 0.00 | 0.00 | 0.00 | 0.00 | 0.00 | 0.00 | 0.00 | 74.00 | 12,184.00 | 2121.00 |
| SRR23425865 | NRB | 4.5M | 9.9M | 0.00 | 0.00 | 0.00 | 0.00 | 0.00 | 0.00 | 0.00 | 0.00 | 0.00 | 0.00 | 47.00 | 18,390.00 | 2630.00 |
| SRR23425866 | NRB | 6.5M | 8.6M | 0.00 | 0.00 | 0.00 | 0.00 | 0.00 | 0.00 | 0.00 | 0.00 | 0.00 | 0.00 | 148.00 | 55,271.00 | 6275.00 |
| SRR23425867 | NRB | 3.7M | 11.7M | 0.00 | 0.00 | 0.00 | 0.00 | 0.00 | 0.00 | 0.00 | 0.00 | 0.00 | 0.00 | 10.00 | 4482.00 | 945.00 |

[a]NRB: no-risk benign.
[b]HRB: high-risk benign.
[c]ESC: early-stage breast cancer.

exogenous EVs (30, 32, 41), and miRNAs within them can regulate gene expression (30). Remarkably, EVs containing miRNAs are resistant to pasteurization, and miRNAs contained in different commercial dairy products (i.e., milk, baby formula milk, etc.) can be absorbed by humans (26, 27, 30, 32, 40). Taken together, these observations allowed us to hypothesize that there could be an association between BLV miRNAs and particular human cancers, even in the absence of BLV exposure.

In this study, we screened 335 publicly available human small-RNA sequencing data sets from BCA, leukemias, and healthy controls to determine whether BLV-encoded miRNAs could be detected in human cancer miRNomes. This hypothesis was motivated by the established oncogenic role of BLV miRNAs in cattle and the ongoing controversy regarding potential BLV exposure, infection, or dietary transfer in humans. Rather than

**TABLE 8** PRJNA792999: tRF and tiRNA sequencing of BCA tissue samples and matched non-tumor adjacent tissues[a]

| Sample ID | Cancer type | Post trim reads | Total no of reads | Number of reads that align on | | | | | | | | | | | | |
|---|---|---|---|---|---|---|---|---|---|---|---|---|---|---|---|---|
| | | | | blv-miR-B1-3p | blv-miR-B1-5p | blv-miR-B2-3p | blv-miR-B2-5p | blv-miR-B3-3p | blv-miR-B3-5p | blv-miR-B4-3p | blv-miR-B4-5p | blv-miR-B5-3p | blv-miR-B5-5p | hsa-miR-106b-5p | hsa-miR-21-5p | hsa-mir-29a-3p |
| SRR17374780 | CONTROL | 3.2M | 7.5M | 0.00 | 0.00 | 0.00 | 0.00 | 0.00 | 0.00 | 0.00 | 0.00 | 0.00 | 0.00 | 846.00 | 511,803.00 | 13,671.00 |
| SRR17374781 | BCA | 5.6M | 8.6M | 0.00 | 0.00 | 0.00 | 0.00 | 0.00 | 0.00 | 0.00 | 0.00 | 0.00 | 0.00 | 1029.00 | 165,760.00 | 21,253.00 |
| SRR17374782 | BCA | 5.6M | 8.6M | 0.00 | 0.00 | 0.00 | 0.00 | 1.00 | 0.00 | 0.00 | 0.00 | 0.00 | 0.00 | 1172.00 | 249,216.00 | 19,424.00 |
| SRR17374783 | CONTROL | 3.5M | 7.4M | 0.00 | 0.00 | 0.00 | 0.00 | 0.00 | 0.00 | 0.00 | 0.00 | 0.00 | 0.00 | 1262.00 | 723,307.00 | 11,318.00 |
| SRR17374784 | CONTROL | 4.1M | 8.5M | 0.00 | 0.00 | 0.00 | 0.00 | 0.00 | 1.00 | 0.00 | 0.00 | 0.00 | 0.00 | 1380.00 | 441,183.00 | 20,845.00 |
| SRR17374785 | CONTROL | 3.4M | 7.0M | 0.00 | 0.00 | 0.00 | 0.00 | 0.00 | 0.00 | 0.00 | 0.00 | 0.00 | 0.00 | 864.00 | 212,775.00 | 10,398.00 |
| SRR17374786 | BCA | 7.7M | 11.4M | 0.00 | 0.00 | 0.00 | 0.00 | 1.00 | 0.00 | 0.00 | 0.00 | 0.00 | 0.00 | 1236.00 | 209,167.00 | 19,403.00 |
| SRR17374787 | CONTROL | 5.3M | 8.0M | 0.00 | 1.00 | 0.00 | 0.00 | 0.00 | 0.00 | 0.00 | 0.00 | 0.00 | 0.00 | 1078.00 | 178,186.00 | 30,266.00 |
| SRR17374788 | BCA | 5.8M | 11.3M | 0.00 | 0.00 | 0.00 | 0.00 | 0.00 | 0.00 | 0.00 | 0.00 | 0.00 | 0.00 | 1150.00 | 161,677.00 | 18,856.00 |
| SRR17374789 | BCA | 7.5M | 11.0M | 0.00 | 0.00 | 0.00 | 0.00 | 0.00 | 0.00 | 0.00 | 0.00 | 0.00 | 0.00 | 1409.00 | 289,567.00 | 27,057.00 |

[a]BCA: breast cancer.

testing causation directly, our objective was to perform a sensitive empirical screen to determine whether any reproducible BLV miRNA signal was detectable in human cancer data sets.

Across 1.98 billion trimmed human reads, only 60 reads aligned permissively to BLV miRNA references, predominantly representing STAR sequences or containing mismatches. These patterns are consistent with expected statistical noise under high-sensitivity alignment conditions rather than genuine biological detection. In contrast, our analysis of BLV-infected cattle data sets robustly detected 9 out of 10 canonical BLV miRNAs, confirming that the analytical workflow can identify authentic BLV miRNAs when present at biologically relevant levels. We therefore interpret the sparse BLV-like alignments in human samples as background noise.

As internal quality controls, we evaluated three human miRNAs linked to oncogenic processes (hsa-miR-29a-3p, hsa-miR-106b-5p, and hsa-miR-21-5p) (21, 42). Their consistent detection across data sets (except PRJNA996975, excluded due to low read depth and impaired recovery of internal controls) demonstrates that the analyzed libraries were suitable for miRNA detection. Together with the high BLV miRNA levels observed in BLV-infected cattle, these results validate both the sensitivity of our workflow and the integrity of the human data sets.

However, several methodological limitations must be acknowledged. First, the data sets analyzed predominantly consist of bulk tumor or peripheral blood sRNA-seq libraries, which may not correspond to biological compartments aligned with proposed exposure models. If the presence of BLV miRNAs in humans arises from direct viral infection, they might localize primarily to lymphoid-associated compartments rather than bulk tumor tissue. In addition, BLV may replicate in cell types distinct from those infected in the natural host. In this regard, Moran and colleagues demonstrated the ability of BLV to infect human mammary epithelial cells (43). Alternatively, if derived through dietary EV transfer, detection may require postprandial plasma sampling, when EV concentrations peak. Thus, our inference applies specifically to the sampled compartments and timepoints represented in the public data sets analyzed. Second, we did not establish quantitative detection limits through matrix-matched synthetic spike-ins processed using the same sequencing workflows. Accordingly, "below detection threshold" reflects the operational sensitivity of this pipeline rather than absolute biological absence. Third, we relied on canonical miRBase BLV references, which may not capture divergent BLV genotypes or non-canonical isomiRs potentially relevant to human exposure. Future analyses incorporating explicit false-positive benchmarking, including competitive decoy alignments and parameter sweeps to quantify mapping specificity under high-sensitivity settings, will help distinguish true low-level signals from alignment noise. Variant-aware mapping and competitive alignment strategies

**TABLE 9** PRJNA758408: Circulating exosomal miRNAs as predictive biomarkers of neoadjuvant chemotherapy response in BCA[a]

| Sample ID | Cancer type | Post trim reads | Total no of reads | Number of reads that align on | | | | | | | | | | | | |
|---|---|---|---|---|---|---|---|---|---|---|---|---|---|---|---|---|
| | | | | blv-miR-B1-3p | blv-miR-B1-5p | blv-miR-B2-3p | blv-miR-B2-5p | blv-miR-B3-3p | blv-miR-B3-5p | blv-miR-B4-3p | blv-miR-B4-5p | blv-miR-B5-3p | blv-miR-B5-5p | hsa-miR-106b-5p | hsa-miR-21-5p | hsa-mir-29a-3p |
| SRR15658006 | BCA | 12.6M | 24.7M | 0.00 | 0.00 | 0.00 | 0.00 | 0.00 | 0.00 | 0.00 | 0.00 | 0.00 | 0.00 | 1.00 | 2281.00 | 954.00 |
| SRR15658007 | BCA | 9.3M | 12.1M | 0.00 | 0.00 | 0.00 | 0.00 | 0.00 | 0.00 | 0.00 | 0.00 | 0.00 | 0.00 | 0.00 | 714.00 | 1239.00 |
| SRR15658008 | BCA | 13.0M | 24.3M | 0.00 | 0.00 | 0.00 | 0.00 | 0.00 | 0.00 | 0.00 | 0.00 | 0.00 | 0.00 | 7.00 | 7433.00 | 2771.00 |
| SRR15658009 | BCA | 14.8M | 25.5M | 0.00 | 2.00 | 0.00 | 0.00 | 0.00 | 0.00 | 0.00 | 0.00 | 0.00 | 0.00 | 50.00 | 49,045.00 | 5029.00 |
| SRR15658010 | BCA | 12.0M | 25.2M | 0.00 | 0.00 | 0.00 | 0.00 | 0.00 | 0.00 | 0.00 | 0.00 | 0.00 | 0.00 | 14.00 | 14,352.00 | 3276.00 |
| SRR15658011 | BCA | 11.8M | 21.5M | 0.00 | 0.00 | 0.00 | 0.00 | 0.00 | 0.00 | 0.00 | 0.00 | 0.00 | 0.00 | 14.00 | 19,317.00 | 5017.00 |
| SRR15658012 | BCA | 13.3M | 24.1M | 0.00 | 0.00 | 0.00 | 0.00 | 0.00 | 0.00 | 0.00 | 5.00 | 0.00 | 0.00 | 0.00 | 3573.00 | 1463.00 |
| SRR15658013 | BCA | 12.9M | 19.1M | 0.00 | 0.00 | 0.00 | 0.00 | 0.00 | 0.00 | 0.00 | 2.00 | 0.00 | 0.00 | 0.00 | 1333.00 | 785.00 |
| SRR15658014 | BCA | 7.3M | 17.1M | 0.00 | 0.00 | 0.00 | 0.00 | 0.00 | 0.00 | 0.00 | 0.00 | 0.00 | 0.00 | 0.00 | 1682.00 | 5269.00 |
| SRR15658015 | BCA | 9.5M | 17.5M | 0.00 | 0.00 | 0.00 | 0.00 | 0.00 | 0.00 | 0.00 | 0.00 | 0.00 | 0.00 | 2.00 | 4631.00 | 8741.00 |
| SRR15658016 | BCA | 8.8M | 20.7M | 0.00 | 0.00 | 0.00 | 0.00 | 0.00 | 0.00 | 0.00 | 0.00 | 0.00 | 0.00 | 6.00 | 6250.00 | 3228.00 |
| SRR15658017 | BCA | 9.7M | 19.9M | 0.00 | 0.00 | 0.00 | 0.00 | 0.00 | 0.00 | 0.00 | 0.00 | 0.00 | 0.00 | 20.00 | 10,140.00 | 3642.00 |
| SRR15658018 | BCA | 10.9M | 20.1M | 0.00 | 0.00 | 0.00 | 0.00 | 0.00 | 0.00 | 0.00 | 0.00 | 0.00 | 0.00 | 59.00 | 38,933.00 | 8483.00 |
| SRR15658019 | BCA | 12.2M | 20.0M | 0.00 | 0.00 | 0.00 | 0.00 | 0.00 | 0.00 | 0.00 | 0.00 | 0.00 | 0.00 | 52.00 | 63,037.00 | 10,393.00 |
| SRR15658020 | BCA | 4.0M | 5.6M | 0.00 | 0.00 | 0.00 | 0.00 | 0.00 | 0.00 | 0.00 | 0.00 | 0.00 | 0.00 | 0.00 | 2473.00 | 2136.00 |
| SRR15658021 | BCA | 16.6M | 23.4M | 0.00 | 0.00 | 0.00 | 0.00 | 0.00 | 0.00 | 0.00 | 0.00 | 0.00 | 0.00 | 14.00 | 5989.00 | 2084.00 |
| SRR15658022 | BCA | 8.5M | 18.0M | 0.00 | 0.00 | 0.00 | 0.00 | 0.00 | 0.00 | 0.00 | 0.00 | 0.00 | 0.00 | 3.00 | 5398.00 | 3700.00 |
| SRR15658023 | BCA | 11.3M | 20.0M | 0.00 | 0.00 | 0.00 | 0.00 | 0.00 | 0.00 | 0.00 | 0.00 | 0.00 | 0.00 | 49.00 | 10,207.00 | 7480.00 |
| SRR15658024 | BCA | 9.5M | 17.3M | 0.00 | 0.00 | 0.00 | 0.00 | 0.00 | 0.00 | 0.00 | 0.00 | 0.00 | 0.00 | 68.00 | 15,731.00 | 6682.00 |
| SRR15658025 | BCA | 11.3M | 20.7M | 0.00 | 0.00 | 0.00 | 0.00 | 0.00 | 0.00 | 0.00 | 0.00 | 0.00 | 0.00 | 41.00 | 12,592.00 | 7573.00 |
| SRR15658026 | BCA | 11.1M | 19.7M | 0.00 | 0.00 | 0.00 | 0.00 | 0.00 | 0.00 | 0.00 | 0.00 | 0.00 | 0.00 | 50.00 | 27,748.00 | 13,484.00 |
| SRR15658027 | BCA | 10.3M | 18.6M | 0.00 | 0.00 | 0.00 | 0.00 | 0.00 | 0.00 | 0.00 | 0.00 | 0.00 | 0.00 | 3.00 | 7453.00 | 5892.00 |
| SRR15658028 | BCA | 11.1M | 19.6M | 0.00 | 0.00 | 0.00 | 0.00 | 0.00 | 0.00 | 0.00 | 11.00 | 0.00 | 0.00 | 34.00 | 21,262.00 | 8018.00 |
| SRR15658029 | BCA | 7.9M | 16.4M | 0.00 | 0.00 | 0.00 | 0.00 | 0.00 | 0.00 | 0.00 | 0.00 | 0.00 | 0.00 | 21.00 | 8610.00 | 4560.00 |
| SRR15658030 | BCA | 8.7M | 17.7M | 0.00 | 0.00 | 0.00 | 0.00 | 0.00 | 0.00 | 0.00 | 0.00 | 0.00 | 0.00 | 48.00 | 7291.00 | 3855.00 |
| SRR15658031 | BCA | 14.2M | 20.3M | 0.00 | 0.00 | 0.00 | 0.00 | 0.00 | 0.00 | 0.00 | 0.00 | 0.00 | 0.00 | 45.00 | 39,215.00 | 21,751.00 |
| SRR15658032 | BCA | 12.4M | 20.4M | 0.00 | 0.00 | 0.00 | 0.00 | 0.00 | 0.00 | 0.00 | 0.00 | 0.00 | 0.00 | 104.00 | 40,141.00 | 11,822.00 |
| SRR15658033 | BCA | 10.1M | 18.9M | 0.00 | 0.00 | 0.00 | 0.00 | 0.00 | 0.00 | 0.00 | 0.00 | 0.00 | 0.00 | 3.00 | 10,767.00 | 9387.00 |
| SRR15658034 | BCA | 10.1M | 19.6M | 0.00 | 0.00 | 0.00 | 0.00 | 0.00 | 0.00 | 0.00 | 0.00 | 0.00 | 0.00 | 18.00 | 23,834.00 | 4825.00 |
| SRR15658035 | BCA | 10.3M | 18.7M | 0.00 | 0.00 | 0.00 | 0.00 | 0.00 | 0.00 | 0.00 | 0.00 | 0.00 | 0.00 | 5.00 | 15,177.00 | 3724.00 |
| SRR15658036 | BCA | 14.3M | 20.8M | 0.00 | 0.00 | 0.00 | 0.00 | 0.00 | 0.00 | 0.00 | 0.00 | 0.00 | 0.00 | 54.00 | 73,731.00 | 24,589.00 |
| SRR15658037 | BCA | 10.5M | 22.2M | 0.00 | 0.00 | 0.00 | 0.00 | 0.00 | 0.00 | 0.00 | 0.00 | 0.00 | 0.00 | 5.00 | 8502.00 | 3850.00 |
| SRR15658038 | BCA | 10.6M | 22.3M | 0.00 | 0.00 | 0.00 | 0.00 | 0.00 | 0.00 | 0.00 | 0.00 | 0.00 | 0.00 | 14.00 | 6727.00 | 3365.00 |
| SRR15658039 | BCA | 11.6M | 21.7M | 0.00 | 0.00 | 0.00 | 0.00 | 0.00 | 0.00 | 0.00 | 0.00 | 0.00 | 0.00 | 22.00 | 8220.00 | 4864.00 |
| SRR15658040 | BCA | 12.1M | 24.4M | 0.00 | 0.00 | 0.00 | 0.00 | 0.00 | 0.00 | 0.00 | 0.00 | 0.00 | 0.00 | 43.00 | 8446.00 | 4863.00 |
| SRR15658041 | BCA | 9.8M | 18.8M | 0.00 | 0.00 | 0.00 | 0.00 | 0.00 | 0.00 | 0.00 | 0.00 | 0.00 | 0.00 | 0.00 | 21,313.00 | 5522.00 |
| SRR15658042 | BCA | 9.1M | 19.6M | 0.00 | 0.00 | 0.00 | 0.00 | 0.00 | 0.00 | 0.00 | 0.00 | 0.00 | 0.00 | 14.00 | 8176.00 | 5296.00 |
| SRR15658043 | BCA | 11.0M | 20.4M | 0.00 | 0.00 | 0.00 | 0.00 | 0.00 | 0.00 | 0.00 | 0.00 | 0.00 | 0.00 | 18.00 | 11,067.00 | 5029.00 |
| SRR15658044 | BCA | 6.1M | 9.5M | 0.00 | 0.00 | 0.00 | 0.00 | 0.00 | 0.00 | 0.00 | 0.00 | 0.00 | 0.00 | 0.00 | 1704.00 | 4615.00 |
| SRR15658045 | BCA | 9.4M | 18.8M | 0.00 | 0.00 | 0.00 | 0.00 | 0.00 | 0.00 | 0.00 | 0.00 | 0.00 | 0.00 | 3.00 | 4907.00 | 5203.00 |

[a]BCA: breast cancer.

against the full human miRNAome/genome may further improve specificity and reduce cross-mapping artifacts, particularly given the shared seed sequences between blv-miR-b4-3p and hsa-miR-29a. Additionally, because the data sets analyzed were not optimized for rare viral miRNA detection, targeted enrichment strategies—such as argonaut immunoprecipitation (AGO-IP), locked nucleic acid (LNA)-based capture probes, or viral hairpin pull-down—may be required during future library preparations to detect extremely low-abundance or transient molecules. AGO-IP enriches for miRNAs actively

**TABLE 10** Number of normalized counts aligned on BLV miRNAs (cattle sRNA-seq)[a]

| BLV status | Animal ID | Total number of reads | NORMALIZED COUNTS that align on (RPM) | | | | | | | | | |
|---|---|---|---|---|---|---|---|---|---|---|---|---|
| | | | blv-miR-B1-3p | blv-miR-B1-5p | blv-miR-B2-5p | blv-miR-B2-3p | blv-miR-B3-5p | blv-miR-B3-3p | blv-miR-B4-3p | blv-miR-B4-5p | blv-miR-B5-5p | blv-miR-B5-3p |
| BLV-infected | SRR5659715 | 14,100,000 | 5224 | 30 | 889 | 1344 | 32 | 314 | 1732 | 0 | 31 | 100 |
| | SRR5659721 | 12,700,000 | 12 | 1 | 3 | 6 | 0 | 2 | 6 | 0 | 0 | 1 |
| | SRR5659722 | 10,700,000 | 313 | 8 | 141 | 255 | 8 | 55 | 113 | 0 | 14 | 33 |
| BLV-uninfected | SRR5659714 | 8,400,000 | 6 | 0 | 1 | 2 | 0 | 0 | 2 | 0 | 0 | 0 |
| | SRR5659718 | 18,700,000 | 10 | 0 | 1 | 3 | 0 | 1 | 4 | 0 | 0 | 0 |
| | SRR5659719 | 17,400,000 | 5 | 0 | 1 | 2 | 0 | 0 | 2 | 0 | 0 | 0 |

[a]RPM: reads per million of mapped reads.

loaded into the RISC complex, whereas LNA-based capture and viral hairpin pull-down selectively enrich predefined viral miRNAs and precursor miRNAs, respectively. Together, these approaches enhance sensitivity for detecting low-abundance or transient viral miRNAs.

In summary, within the limitations of publicly available small-RNA data sets, canonical reference alignments, and standard sequencing workflows, we found no evidence supporting the presence of BLV miRNAs in human BCA and human leukemia samples. While trace or transient signals cannot be entirely excluded, the data indicate that BLV miRNAs are not a reproducible feature of human cancer miRNomes. These findings contribute empirical clarity to a longstanding controversy and highlight the need for future research incorporating targeted enrichment, spike-in calibration, variant-aware reference sets, and compartment- or time-specific sampling to definitively evaluate mechanistic hypotheses of BLV exposure in humans.

## ACKNOWLEDGMENTS

The author thanks Dr. Carignano for past discussions related to the broader research topic. However, the present study, including the conception, data collection, analysis, and manuscript preparation, was conducted independently by the author. The author acknowledges the Instituto de Virología e Innovaciones Tecnológicas (IVIT), Instituto Nacional de Tecnología Agropecuaria (INTA), and Consejo Nacional de Investigaciones Científicas y Técnicas (CONICET) for supporting the investigation. The author also thanks OpenAI's ChatGPT for assisting with manuscript preparation, including suggestions for language editing and improving clarity.

J.P.J. was involved in the study conception, experimental design, execution, analysis, interpretation of data, writing, and funding acquisition.

## AUTHOR AFFILIATION

[1]Instituto de Virología e Innovaciones Tecnológicas (IVIT), Instituto Nacional de Tecnología Agropecuaria (INTA), Consejo Nacional de Investigaciones Científicas y Tecnológicas (CONICET), Buenos Aires, Argentina

## AUTHOR ORCIDs

Juan Pablo Jaworski  http://orcid.org/0000-0002-1311-060X

## FUNDING

| Funder | Grant(s) | Author(s) |
|---|---|---|
| Fondo para la Investigación Científica y Tecnológica | PICT2017-0262 | Juan Pablo Jaworski |

## AUTHOR CONTRIBUTIONS

Juan Pablo Jaworski, Conceptualization, Data curation, Formal analysis, Funding acquisition, Investigation, Methodology, Project administration, Resources, Supervision, Validation, Writing – original draft, Writing – review and editing

## ETHICS APPROVAL

The study uses publicly available, de-identified data from SRA (NCBI).

## ADDITIONAL FILES

The following material is available online.

### Supplemental Material

**Figure S1 (Spectrum03818-25-s0001.docx).** Alignment of sRNA seq reads from human cancer sample SRR15658028 (PRJNA758408) on BLV miRNAs.
**Table S1 (Spectrum03818-25-s0002.docx).** Reference miRNA.

### Open Peer Review

**PEER REVIEW HISTORY (review-history.pdf).** An accounting of the reviewer comments and feedback.

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
