## [Reviewer comments · Microbiology Spectrum]

Microbiology Spectrum

Absence of detectable bovine leukemia virus miRNAs in human cancer small RNA-seq datasets

Juan Jaworski

Corresponding Author(s): Juan Jaworski, Instituto Nacional de Tecnologia Agropecuaria

Review Timeline:

Submission Date:	November 25, 2025
Editorial Decision:	January 5, 2026
Revision Received:	January 7, 2026
Accepted:	January 8, 2026

Editor: Peter Pelka

Reviewer(s): The reviewers have opted to remain anonymous.

Transaction Report:

DOI: <https://doi.org/10.1128/spectrum.03818-25>

Re: Spectrum03818-25 (Lack of association between bovine leukemia virus miRNAs and human leukemia or breast cancer)

Dear Dr. Juan Pablo Jaworski:

Thank you for the privilege of reviewing your work. Below you will find my comments, instructions from the Spectrum editorial office, and the reviewer comments.

Thank you for submitting your manuscript to Spectrum. The reviewers found the manuscript of interest and have suggested some small changes as outlined below in their comments.

Revision Guidelines

Sincerely,
Peter Pelka
Editor
Microbiology Spectrum

Reviewer #1 (Comments for the Author):

This study conducted a highly sensitive screening for bovine leukemia virus (BLV)-derived miRNAs using publicly available NGS datasets obtained from various human cancers, including breast cancer, acute lymphoblastic leukemia, acute myeloid leukemia, and chronic lymphocytic leukemia. The analysis found no BLV-derived miRNA sequences in these human datasets; however,

human miRNAs such as miR-29a-3p and miR-106-5p were clearly detected. The workflow was validated by successfully identifying BLV-derived miRNAs in NGS data obtained from BLV-infected animals. In summary, the results provided no evidence of BLV presence in human cancer cases.

As noted by the author, the potential link between BLV and human cancer remains a subject of debate. The objective of this study was to determine whether any detectable presence of BLV miRNAs exists in human cancer, which may help address this ongoing controversy. The study's rationale is well-defined, and the conclusions are supported by the results. My minor comments are described below.

P1L2 Title: The term "Lack of association" is not suitable, as the purpose of this study is to determine whether BLV miRNA signals can be detected across multiple datasets derived from human cancer cases, rather than to investigate any association between BLV and human cancer.

P5L102: The reference file described here likely refers to the self-assembled miRNA sequences shown in Supplementary table 1, not the bovine genome.

P6L128: "BLV" has already been written in full in L38.

P9L193: Please spell out AGO-IP, LNA, and viral hairpin pull-down.

Reviewer #2 (Comments for the Author):

Dear author: after reading your manuscript entitled "Lack of association between bovine leukemia virus miRNAs and human leukemia or breast cancer" there are some issues I would like to discuss.

In your work, you attempted to detect sequences of BLV miRNA in different data bases from human small RNA-seq datasets, including different types of cancers and normal tissues. You were not able to detect any BLV sequence, regardless of some scarce detection that probably could be considered background noise.

Title: Instead of "Lack of association...." I would recommend changing the title to something like "Undetectable BLV miRNA in different databases derived from human tumors" or another option more closely aligned with the purpose of the manuscript. The current title, in my opinion, does not accurately describe the results you have obtained. You were not investigating an association, rather, you were attempting to detect the presence of the miRNAs.

It reminds me to a paper from Chinese authors "Lack of association between bovine leukemia virus and breast cancer in Chinese patients" published in 2016.

Lane 130: BLV miRNAs were detected at lower frequencies in 130 BLV-uninfected cattle and were attributed to background noise or early infection stages.

As it has been described repeatedly, to confirm that a bovine is infected with BLV a second determination must be performed at least three months later after the first screening.

Lane 150: Taken together, these observations allowed us to hypothesize that there could be an association between BLV miRNAs and particular human cancers, even in the absence of BLV exposure.

What do you mean by "absence of BLV exposure"? In such a case, the probability of becoming infected is almost null.

Lane 178: If BLV miRNAs arise from direct viral infection, they might localize primarily to B-cell and lymphoid microenvironments rather than bulk tumor.

The human BLV receptor has not yet been accurately described, and there is no evidence that the host cells are B lymphocytes, as in bovines, nor that BLV infects tumor cells other than cell mammary epithelial cells (Moran PE et al, 2025). Please revise this assertion.

Table 2. PRJNA996975: Profiling of miRNA in pediatric Acute Myeloid Leukemia (AML) in Egyptian population

The presence of BLV-whether proviral fragments, antibodies, or any other indication of infection-has never been described in children. BLV is a disease with a slow progression that takes many years to become established; therefore, it is highly unlikely that any signal would be detected in pediatric datasets.

Table 9. PRJNA758408: Circulating exosomal microRNAs as predictive biomarkers of neoadjuvant chemotherapy response in breast cancer

I don't believe that this search is relevant to the aims of the manuscript.

Overall, It appears that you had access to several bioinformatic tools and explored databases focused on human microRNAs, which were then used to attempt the detection of BLV-derived microRNAs, without a comprehensive understanding of viral pathogenesis. Nevertheless, the work is commendable, despite its multiple limitations, as you yourself acknowledge.

I also have a comment about the tables. In addition, I have a comment regarding the tables. The manuscript would be much easier to read if, instead of presenting those large tables with raw results, a concise summary of the findings were provided. I find it cumbersome to review a manuscript containing extensive, detailed tables whose results have already been discussed in

the main text. I therefore suggest summarizing these tables and, if necessary, moving the full versions to the supplementary material for readers who may wish to consult them.

RESPONSE TO REVIEWERS

I would like to thank the Reviewers for their helpful comments. Below you will find point-by-point responses to the issues raised by the Reviewers. Reviewer comments are shown in **BOLD**, followed by the author's responses in plain text.

Reviewer #1:

This study conducted a highly sensitive screening for bovine leukemia virus (BLV)-derived miRNAs using publicly available NGS datasets obtained from various human cancers, including breast cancer, acute lymphoblastic leukemia, acute myeloid leukemia, and chronic lymphocytic leukemia. The analysis found no BLV-derived miRNA sequences in these human datasets; however, human miRNAs such as miR-29a-3p and miR-106-5p were clearly detected. The workflow was validated by successfully identifying BLV-derived miRNAs in NGS data obtained from BLV-infected animals. In summary, the results provided no evidence of BLV presence in human cancer cases. As noted by the author, the potential link between BLV and human cancer remains a subject of debate. The objective of this study was to determine whether any detectable presence of BLV miRNAs exists in human cancer, which may help address this ongoing controversy. The study's rationale is well-defined, and the conclusions are supported by the results. My minor comments are described below.

P1L2 Title: The term "Lack of association" is not suitable, as the purpose of this study is to determine whether BLV miRNA signals can be detected across multiple datasets derived from human cancer cases, rather than to investigate any association between BLV and human cancer.

Response: I would like to thank the Reviewer for this recommendation; I modified the title of the manuscript to:

"Absence of detectable bovine leukemia virus miRNAs in human cancer small RNA-seq datasets"
(LINE 1)

P5L102: The reference file described here likely refers to the self-assembled miRNA sequences shown in Supplementary table 1, not the bovine genome.

Response: I appreciate this observation from the Reviewer. I noticed this wording error in the Methods section after submission, and it has now been corrected. The miRDeep2 mapper was used only to collapse reads prior to miRDeep2 quantifier, and no genome alignment step was conducted in the context of BLV miRNA detection. The corrected description now reads:

"Reads were collapsed using the miRDeep2 mapper tool".

(LINE 107)

This correction has no effect on the results, interpretation, or conclusions.

P6L128: "BLV" has already been written in full in L38.

Response: Corrected to "BLV". (LINE 132)

P9L193: Please spell out AGO-IP, LNA, and viral hairpin pull-down.

Response: The text now reads:

"Additionally, because the datasets analysed were not optimized for rare viral miRNA detection, targeted enrichment strategies—such as argonaut immunoprecipitation (AGO-IP), locked nucleic acid (LNA)-based capture probes, or viral hairpin pull-down—may be required during future library preparations to detect extremely low-abundance or transient molecules. AGO-IP enriches for miRNAs actively loaded into the RISC complex, whereas LNA-based capture and viral hairpin pull-down selectively enrich predefined viral miRNAs and precursors miRNAs, respectively. Together, these approaches enhance sensitivity for detecting low-abundance or transient viral miRNAs."

(LINE 197-203)

Reviewer #2:

Dear author: after reading your manuscript entitled "Lack of association between bovine leukemia virus miRNAs and human leukemia or breast cancer" there are some issues I would like to discuss. In your work, you attempted to detect sequences of BLV miRNA in different data bases from human small RNA-seq datasets, including different types of cancers and normal tissues. You were not able to detect any BLV sequence, regardless of some scarce detection that probably could be considered background noise.

Title: Instead of "Lack of association...." I would recommend changing the title to something like "Undetectable BLV miRNA in different databases derived from human tumors" or another option more closely aligned with the purpose of the manuscript. The current title, in my opinion, does not accurately describe the results you have obtained. You were not investigating an association, rather, you were attempting to detect the presence of the miRNAs.

It reminds me to a paper from Chinese authors "Lack of association between bovine leukemia virus and breast cancer in Chinese patients" published in 2016.

Response: I would like to thank the Reviewer for this recommendation; I modified the title of the manuscript to:

"Absence of detectable bovine leukemia virus miRNAs in human cancer small RNA-seq datasets"

Lane 130: BLV miRNAs were detected at lower frequencies in BLV-uninfected cattle and were attributed to background noise or early infection stages.

As it has been described repeatedly, to confirm that a bovine is infected with BLV a second determination must be performed at least three months later after the first screening.

Response: As explained in the Methods section, six animals (BLV-positive; N=3 and BLV-negative; N=3) used to validate the analytical workflow were derived from a previously published study not conducted by our laboratory (REF: Casas E, Ma H, Lippolis JD. 2020. Expression of Viral microRNAs in Serum and White Blood Cells of Cows Exposed to Bovine Leukemia Virus. Front Vet Sci 7:677); therefore, we cannot independently confirm their infection status.

The presence of BLV miRNAs in control animals could be attributed to several factors. First, the original study did not include acute BLV infection status as part of the inclusion criteria, potentially allowing animals in early stages of infection to be categorized as BLV-negative. Second, the detected miRNAs might represent exogenous miRNAs or sequences, unrelated to BLV exposure. Lastly, we cannot exclude the possibility that the low RPM values observed in BLV-negative samples reflect background noise inherent to the data analysis process.

Lane 150: Taken together, these observations allowed us to hypothesize that there could be an association between BLV miRNAs and particular human cancers, even in the absence of BLV exposure.

What do you mean by "absence of BLV exposure"? In such a case, the probability of becoming infected is almost null.

Response: I thank the Reviewer for this important comment. As stated in the final paragraph of the Introduction and in the Discussion (lines 161-162), the objective of the present study was to perform a sensitive empirical assessment of the presence of BLV miRNAs in human cancer datasets, rather than to test causation. BLV infection in humans and its potential role in human cancer remain controversial. Following this line of investigation, we explored an alternative viral factor potentially linked to oncogenesis: BLV miRNAs. Viral miRNAs can exert oncogenic effects independently; for example, BLV miR-b4-3p is functionally analogous to well characterized oncomiR miR-29a. We hypothesized that BLV miRNAs could appear in human subjects either as a consequence of BLV exposure or through the consumption of cattle-derived products containing BLV miRNAs, even in the absence of productive viral infection.

Lane 178: If BLV miRNAs arise from direct viral infection, they might localize primarily to B-cell and lymphoid microenvironments rather than bulk tumor.

The human BLV receptor has not yet been accurately described, and there is no evidence that the host cells are B lymphocytes, as in bovines, nor that BLV infects tumor cells other than cell mammary epithelial cells (Moran PE et al, 2025). Please revise this assertion.

Response: I thank the Reviewer for this insightful comment. We have revised the text accordingly. The corrected passage now reads:

“If the presence of BLV miRNAs in humans arises from direct viral infection, they might localize primarily to lymphoid-associated compartments rather than bulk tumor tissue. In addition, BLV may replicate in cell types distinct from those infected in the natural host. In this regard, Moran and colleagues demonstrated the ability of BLV to infect human mammary epithelial cells.”

LINE 181-185; reference Moran 2025 (#43) added to Reference section of the current manuscript.

Table 2. PRJNA996975: Profiling of miRNA in pediatric Acute Myeloid Leukemia (AML) in Egyptian population. The presence of BLV-whether proviral fragments, antibodies, or any other indication of infection-has never been described in children. BLV is a disease with a slow progression that takes many years to become established; therefore, it is highly unlikely that any signal would be detected in pediatric datasets. Table 9. PRJNA758408: Circulating exosomal microRNAs as predictive biomarkers of neoadjuvant chemotherapy response in breast cancer. I don't believe that this search is relevant to the aims of the manuscript. Overall, It appears that you had access to several bioinformatic tools and explored databases focused on human microRNAs, which were then used to attempt the detection of BLV-derived microRNAs, without a comprehensive understanding of viral pathogenesis. Nevertheless, the work is commendable, despite its multiple limitations, as you yourself acknowledge.

Response: we appreciate the Reviewer's insights regarding the limitations of the present study. Dataset PRJNA996975 was excluded from the final analysis due to low sequencing depth and poor recovery of internal controls, as described in the Results (line 129-130) and Discussion (line 174). This dataset was retained to illustrate the importance of internal controls in evaluating dataset integrity (see line 176). Limitations associated with the use of publicly available datasets are further discussed in lines 179 and 205 of the Discussion.

I also have a comment about the tables. In addition, I have a comment regarding the tables. The manuscript would be much easier to read if, instead of presenting those large tables with raw results, a concise summary of the findings were provided. I find it cumbersome to review a manuscript containing extensive, detailed tables whose results have already been discussed in the main text. I therefore suggest summarizing these tables and, if necessary, moving the full versions to the supplementary material for readers who may wish to consult them.

Response: I thank the Reviewer for this suggestion. I chose to retain the full tables in the main manuscript to ensure transparency and allow readers to directly inspect all the results, giving the central importance of distinguishing true signals from noise.

Re: Spectrum03818-25R1 (Absence of detectable bovine leukemia virus miRNAs in human cancer small RNA-seq datasets)

Dear Dr. Juan Pablo Jaworski:

Your manuscript has been accepted, and I am forwarding it to the ASM production staff for publication. Your paper will first be checked to make sure all elements meet the technical requirements. ASM staff will contact you if anything needs to be revised before copyediting and production can begin. Otherwise, you will be notified when your proofs are ready to be viewed.

Sincerely,
Peter Pelka
Editor
Microbiology Spectrum